# Integrated microwave photonic notch filter using a heterogeneously integrated Brillouin and active-silicon photonic circuit

Matthew Garrett [●][1,2], Yang Liu [●][1,2], Moritz Merklein [●][1,2] [✉], Cong Tinh Bui[1,2], Choon Kong Lai[1,2], Duk-Yong Choi [●][3], Stephen J. Madden[3], Alvaro Casas-Bedoya[1,2] & Benjamin J. Eggleton [●][1,2] [✉]

Microwave photonics (MWP) has unlocked a new paradigm for Radio Frequency (RF) signal processing by harnessing the inherent broadband and tunable nature of photonic components. Despite numerous efforts made to implement integrated MWP filters, a key RF processing functionality, it remains a long-standing challenge to achieve a fully integrated photonic circuit that can merge the megahertz-level spectral resolution required for RF applications with key electro-optic components. Here, we overcome this challenge by introducing a compact 5 mm × 5 mm chip-scale MWP filter with active E-O components, demonstrating 37 MHz spectral resolution. We achieved this device by heterogeneously integrating chalcogenide waveguides, which provide Brillouin gain, in a complementary metal-oxide-semiconductor (CMOS) foundry-manufactured silicon photonic chip containing integrated modulators and photodetectors. This work paves the way towards a new generation of compact, high-resolution RF photonic filters with wideband frequency tunability demanded by future applications, such as air and spaceborne RF communication payloads.

Radio-frequency (RF) filters play a critical role in maintaining signal fidelity of RF and microwave systems by eliminating unwanted signals and preserving desired signals of interest[1]. RF filters require fine spectral resolution to operate within crowded RF spectral environments and must also exhibit broadband frequency tunability as operating frequencies of RF systems move toward millimetre wave bands above 30 GHz[2]. At the same time, they must also achieve small size, weight, and power (SWaP) to meet the demands for operation in future RF applications. However, state-of-the-art RF filter technology fails to simultaneously meet these requirements[3]. Common RF filter implementations based on surface acoustic waves (SAW) are compact, but face challenges operating at high frequencies. For example SAW filters have large losses above 3.5 GHz[4,5], whereas filters based on bulk

acoustic waves can operate into the X-band (8–12 GHz), yet they face fabrication challenges to extend into millimetre wave bands[5,6]. Yttrium iron garnet (YIG) filters are widely tunable, however their form factor is too large for deployment in SWaP-constrained applications[7,8]. Another approach is to use switched filter banks which offer compact form factors, although their frequency response cannot be continuously tuned[9].

Microwave photonic (MWP) filters overcome the frequency tunability limitations of traditional RF filtering technologies by utilising the broadband tunable and reconfigurable nature of optical components. Moreover, with advances in photonic integration, MWP processors that are fully integrated have the potential to offer the compact footprints required for next-generation microwave

[1]Institute of Photonics and Optical Science (IPOS), School of Physics, The University of Sydney, Sydney, NSW 2006, Australia. [2]The University of Sydney Nano Institute (Sydney Nano), The University of Sydney, Sydney, NSW 2006, Australia. [3]Laser Physics Centre, Department of Quantum Science and Technology, Research School of Physics, Australian National University, Canberra, ACT 2601, Australia. [✉]e-mail: moritz.merklein@sydney.edu.au; benjamin.eggleton@sydney.edu.au

systems[1,2,10–12]. Although significant progress on integrated MWP filters has been made, they face difficulties in achieving the fine spectral resolution that is indispensable in RF communication and sensing applications[1]. One of the most common methods to implement on-chip MWP filters is to use integrated optical micro-ring resonators (MRRs), which exhibit compact form factor, as well as reconfigurable amplitude and phase responses[1,13]. However, the achievable 3-dB bandwidth of MRR-based MWP filters is typically limited to hundreds of MHz due to waveguide losses[10,11,14]. Although ultra-high-quality factor (Q) MRRs can be fabricated, they are extremely sensitive to input signal power and temperature fluctuations due to thermo-optic effects[15,16], which can add challenges in aligning the laser to the MRR resonance. Additionally, the free-spectral range (FSR) of the MRR can limit the frequency operating range of the resultant MWP processor. MWP filters based on optical tapped delay lines offer highly reconfigurable RF responses, yet are limited in spectral resolution to GHz levels as they require complex amplitude and phase control of many optical filter taps[17–19]. For example, Shu et al. demonstrated an MWP filter that generated an optical comb and processed each comb line on a photonic chip, but due to the limited number of optical comb lines, the filter 3-dB bandwidth was limited to 900 MHz[20]. Similar to MRR-based processors, the periodic response of tapped delay line MWP processors can limit their frequency operating range.

Stimulated Brillouin scattering (SBS) is an optical nonlinearity first introduced in 1922[21] that has since re-gained interest for use in integrated MWP filters with fine spectral resolution following the first demonstration of SBS in a photonic chip in 2011[22]. SBS is facilitated through a third-order nonlinear acousto-optic interaction[2,21,23,24] and exhibits a narrow intrinsic optical resonant linewidth (10's MHz) due to the long phonon lifetime. In addition to fine spectral resolution, SBS offers several advantages for MWP processing applications as the Brillouin response does not exhibit an FSR, which does not impose any limitations on the operating range of the resultant MWP system[1,2]. Furthermore, the thermal stability of the Brillouin frequency shift (BFS), $\approx 1\,\mathrm{MHz/°C}$ in optical fibres[25], can be much greater than the thermal stability of MRRs, for example, a silicon MRR with Q of $10^7$ exhibited a thermal sensitivity of $\approx 12\,\mathrm{GHz/°C}$[16]. This highlights the advantages of SBS for MWP processing in terms of tunability and stability.

SBS has been generated in a number of chip-scale platforms[22,23,26–30] which enable critical chip-scale MWP processing systems, such as high-performance MWP filters[30–36], phase shifters[37,38], true time delays[39,40], frequency converters[41,42], and high spectral purity microwave sources[43,44]. Silicon photonic platforms, which have the advantage of being compatible with complementary metal-oxide-semiconductor (CMOS) processes and incorporate an extensive passive and active component library, exhibit large Brillouin gain coefficients[23], which is highly desirable for Brillouin-based MWP platforms. Despite these advantages, silicon nanowires on oxide lack intrinsic acoustic guidance, consequently requiring suspended or under-etched structures and exhibit nonlinear optical losses that limit the achievable on-chip Brillouin gain at optical telecommunication wavelengths. On the other hand, silicon nitride ($Si_3N_4$) has negligible nonlinear loss, but only offers minuscule SBS gain coefficients[30,45] due to the modest elasto-optic coefficient that is intrinsic to $Si_3N_4$[45]. Conversely, soft $As_2S_3$ chalcogenide glasses offer large Brillouin gain coefficients[22] and facilitate more than 30 dB of SBS gain in a compact footprint[46], owing to the large elasto-optic coefficient[47] and simultaneous confinement of acoustic and optical modes[22,46,48]. However, chalcogenide waveguides can only be used as a standalone element in MWP filter systems with key MWP components, such as electro-optic (E-O) modulators and photodetectors, remaining off-chip. Previously, we introduced a platform that heterogeneously integrated chalcogenide Brillouin circuits with passive silicon waveguides on a CMOS-compatible SOI platform[49]. However, the passive platform did not take

full advantage of integration with silicon photonics because it lacked an E-O interface. To reduce the SWaP of Brillouin-based MWP processors, integrating chalcogenide Brillouin circuits with the required components of an MWP link, including E-O modulators and photodetectors (PDs), is essential[23].

In this paper, we heterogeneously integrate $As_2S_3$ Brillouin waveguides in a silicon photonic platform, that includes active E-O modulators and photodetectors, to form the basis of a compact MWP processing platform with fine spectral resolution. We demonstrate an integrated MWP notch filter using on-chip silicon devices and Brillouin gain in $As_2S_3$ waveguides for fine spectral resolution processing, achieving 51 dB out-of-band rejection with narrow 37 MHz 3-dB bandwidth and tunable notch central frequencies over 15 GHz. Our demonstration paves the way towards a fully integrated Brillouin-based MWP processor with a reduced footprint and advanced performance.

## Results

A schematic of the heterogeneously integrated MWP notch filter with fine spectral resolution is exemplified in Fig. 1a. An RF signal consisting of two tones with frequencies $f_1$ and $f_2$ is input into the MWP filter. After filtering, one tone is attenuated, as indicated by the spectrally selective notch response. The key components of the filter, including E-O modulator, $As_2S_3$ Brillouin waveguide and photodetector, are implemented in the single compact 5 mm × 5 mm silicon photonic chip presented in Fig. 1b. We exemplify the future feasibility of achieving a fully-integrated MWP system on a chip by incorporating a design that facilitates lasers through the micro-transfer-print technique[50]. However, the lasers were not populated in the current demonstration. We designed a custom E-O modulator, Fig. 1c, that transduces the RF signal into the optical domain and enables flexible control of the optical carrier-sideband ratio (CSR) and sideband phases. The chalcogenide Brillouin waveguide that facilitates fine-spectral resolution processing is highlighted in Fig. 1d.

### Heterogeneous integration of Brillouin $As_2S_3$ waveguides in foundry-manufactured Si photonic circuit

Figure 2 presents the details of the $As_2S_3$ waveguides that, when integrated into the active silicon photonic platform using custom post-processing steps outlined below, enhance the standard silicon photonic chips with nonlinear Brillouin processing capabilities. The $As_2S_3$ Brillouin waveguides lie in a trench defined by the IMEC process flow, where the back-end-of-line stack—including metal layers that would otherwise prevent access to silicon waveguides—was removed. Before the $As_2S_3$ film can be deposited, a 2 μm silica layer covering the silicon waveguides was removed using hydrofluoric acid (HF) etching. Figure 2a depicts the exposed Si waveguide and trench sidewall after HF etching. Subsequently, thermal evaporation was used to deposit a 680 nm thick $As_2S_3$ film and 1800 nm wide waveguides were patterned using electron beam lithography. The Si-$As_2S_3$ taper after resist patterning is shown in Fig. 2b. Finally, a rib waveguide was formed by etching the film to a depth of 340 nm (50%) using inductively coupled plasma-reactive ion etching (RIE). The silicon waveguide tip at the end of the taper is shown in Fig. 2c and a cross-section of the $As_2S_3$ rib waveguide is shown in Fig. 2d. A photograph of the waveguides after fabrication is shown in Fig. 2e. The spiral waveguides were designed with Euler bends using an effective radius of 100 μm and minimal bend losses were observed (See discussion in Supplementary Information section IA). Although the $As_2S_3$ post-processing steps used in this work are similar to the passive Si-$As_2S_3$ platform reported by ref. 49, the advantage of integrating $As_2S_3$ into this active platform comes at the expense of increased fabrication complexity associated with the HF etching step.

We measured $As_2S_3$ propagation losses of 2.0 dB/cm in the Brillouin spiral waveguides using optical frequency domain

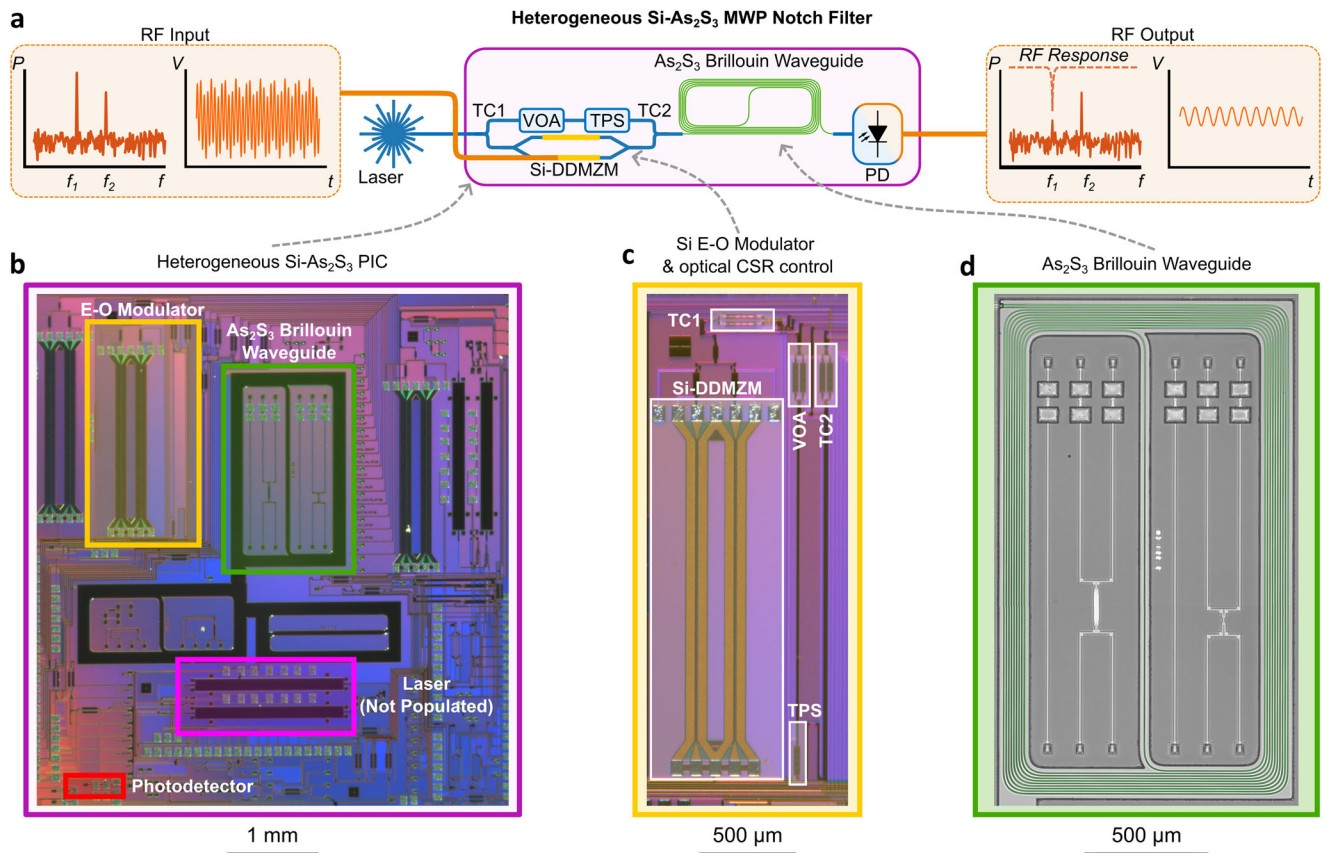

**Fig. 1 | Heterogeneous integration of As$_2$S$_3$ waveguides and silicon components for an on-chip microwave photonic filter system. a** High-level schematic of the integrated Si-As$_2$S$_3$ MWP notch filter. An RF input signal is modulated onto an optical carrier by a modulator on the Si-As$_2$S$_3$ chip. The modulated optical signal is then processed and undergoes photodetection. The notch filter response is visualised in the RF output spectrum where the spectral component at $f_1$ is attenuated, whereas the signal at $f_2$ is not. **b** A picture of the Si-As$_2$S$_3$ PIC containing (**c**) a silicon E-O modulator and circuits used for optical carrier control and (**d**) False-coloured optical microscope image of As$_2$S$_3$ spiral waveguides that are used for Brillouin processing. PIC Photonic integrated circuit, E-O Electro-optic, CSR Carrier-sideband-ratio, TC Tunable coupler, TPS Tunable phase shifter, VOA Variable optical attenuator.

reflectometry, which is significantly higher than the loss of 0.7 dB/cm measured by Morrison et al. on a passive Si-As$_2$S$_3$ platform[49]. To investigate the source of increased propagation losses, we fabricated standalone As$_2$S$_3$ chips (50% etched and fully-etched) with the same waveguide dimensions and spiral layout as fabricated on the Si-As$_2$S$_3$ chip. One can note that the only difference in the fabrication process between the standalone As$_2$S$_3$ chips and the Si-As$_2$S$_3$ chips was that the standalone chips were not exposed to HF before As$_2$S$_3$ deposition. The standalone chip exhibited propagation losses of only 0.4 dB/cm for both etch depths. This is much lower than the 2.0 dB/cm for the 50% etched and 3.2 dB/cm for the fully-etched waveguides on the Si-As$_2$S$_3$ platform (See Supplementary Information Fig. S2–S4).

Scattering from waveguide sidewalls, as well as surface roughness induced from the HF etching step could have increased the losses of the As$_2$S$_3$ waveguides on the heterogeneous platform. This is evidenced by the reduced As$_2$S$_3$ propagation losses on the standalone As$_2$S$_3$ platform that was not exposed to HF before chalcogenide deposition, as well as the increased losses of the fully-etched waveguides compared to the 50% etched waveguides on the Si-As$_2$S$_3$ platform. We measured the root mean squared (RMS) roughness of the SiO$_2$ substrate on the heterogeneous platform before and after HF etching using atomic force microscopy (AFM). The RMS roughness increased from 0.3 nm to 3.2 nm after HF etching. The increased SiO$_2$ roughness, in turn, affects the surface roughness of the As$_2$S$_3$ film that is deposited in the Si-As$_2$S$_3$ platform. This can lead to increased sidewall roughness once the waveguides undergo RIE etching and can

explain why the rib waveguide exhibits significantly reduced propagation losses compared to the fully-etched waveguide on the Si-As$_2$S$_3$ platform (See discussion in Supplementary Information section IA and Fig. S4).

To adiabatically transition the optical signal into the As$_2$S$_3$ waveguide, we taper the width of the silicon waveguide from 450 nm to a minimum tip width of 150 nm whilst maintaining 220 nm height. The TE$_{00}$ mode of the silicon waveguide is input to the taper and the output of the taper is the TE$_{00}$ mode of the As$_2$S$_3$ rib waveguide, which is 1800 nm wide, 680 nm tall with a 340 nm etch. The electric field of the TE$_{00}$ mode of the Si and As$_2$S$_3$ waveguides are plotted in Fig. 2f, g, respectively. We ensured that light was coupled to the TE$_{00}$ mode of the silicon waveguide by adjusting the polarisation of the input light into the chip until the transmitted power was maximised. The simulated optical propagation through the silicon taper is shown in Fig. 2h, which indicates a loss of 0.39 dB per Si-As$_2$S$_3$ interface (See Supplementary Information Fig. S6). We measured losses between 1.8 and 5.5 dB per Si-As$_2$S$_3$ interface. Both are significantly higher than the eigenmode expansion simulation we performed and the 0.1 dB reported by ref. 49.

We measured the magnitude response of the Brillouin resonance in a 5.55 cm As$_2$S$_3$ waveguide on the Si-As$_2$S$_3$ platform with the setup detailed in Fig. 3a (see Methods). Backwards SBS occurs in As$_2$S$_3$ waveguides when an optical signal with large intensity (pump) is counter-propagated against a weaker signal (probe). The pump perturbs the waveguide through electrostriction and the photoelastic effect[23], forming an acoustic wave. When the pump and probe are

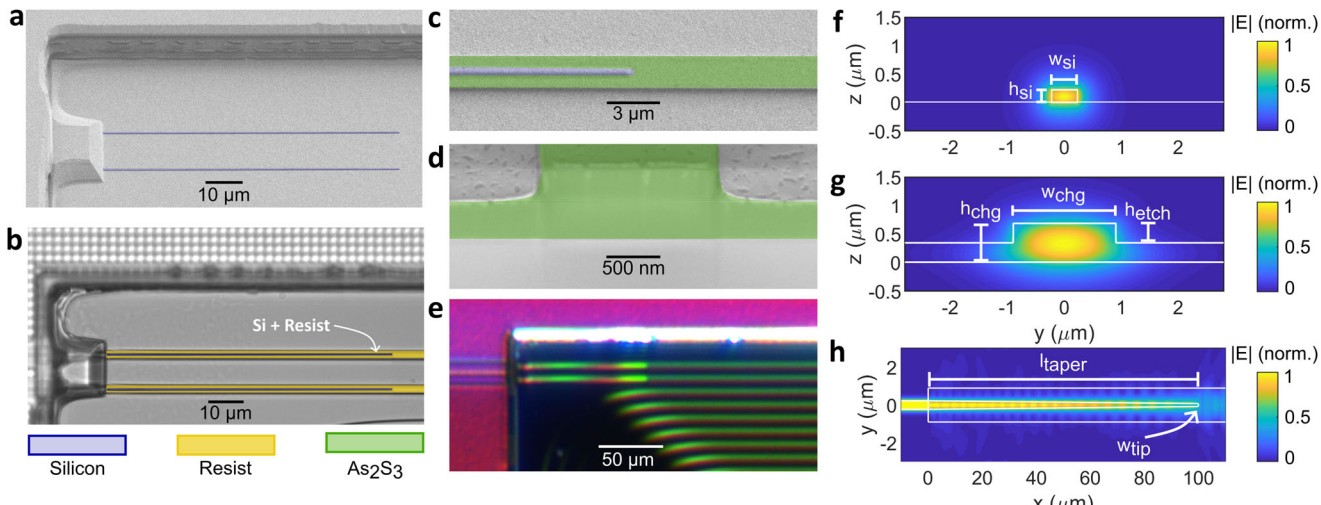

**Fig. 2 | As₂S₃ waveguide fabrication and Si-As₂S₃ interface simulation. a** False-coloured scanning electron microscope (SEM) image of the silicon waveguide taper after the SiO₂ cladding was removed using HF etching. **b** False-coloured optical microscope image of the silicon taper after photoresist is patterned. **c** False-coloured SEM image of the Si waveguide tip and As₂S₃ waveguide. **d** False-coloured SEM image of the As₂S₃ rib waveguide cross-section. **e** True-coloured photograph of the Si-As₂S₃ taper region after fabrication is complete. **f** Simulated electric field magnitude ($|E|$) of the $TE_{00}$ mode profile for the silicon waveguide. **g** Simulated $|E|$ of the $TE_{00}$ mode profile for the As₂S₃ rib waveguide. The mode profiles were simulated using Lumerical Finite Difference Eigenmode (FDE) solver. **h** Simulated $|E|$ through the Si taper into the As₂S₃ waveguide, simulated using Lumerical Eigenmode Expansion (EME) solver. The labelled critical dimensions have values $w_{Si} = 450$ nm, $h_{Si} = 220$ nm, $l_{taper} = 100$ μm, $w_{tip} = 150$ nm, $h_{chg} = 680$ nm, $h_{etch} = 340$ nm, $w_{chg} = 1800$ nm.

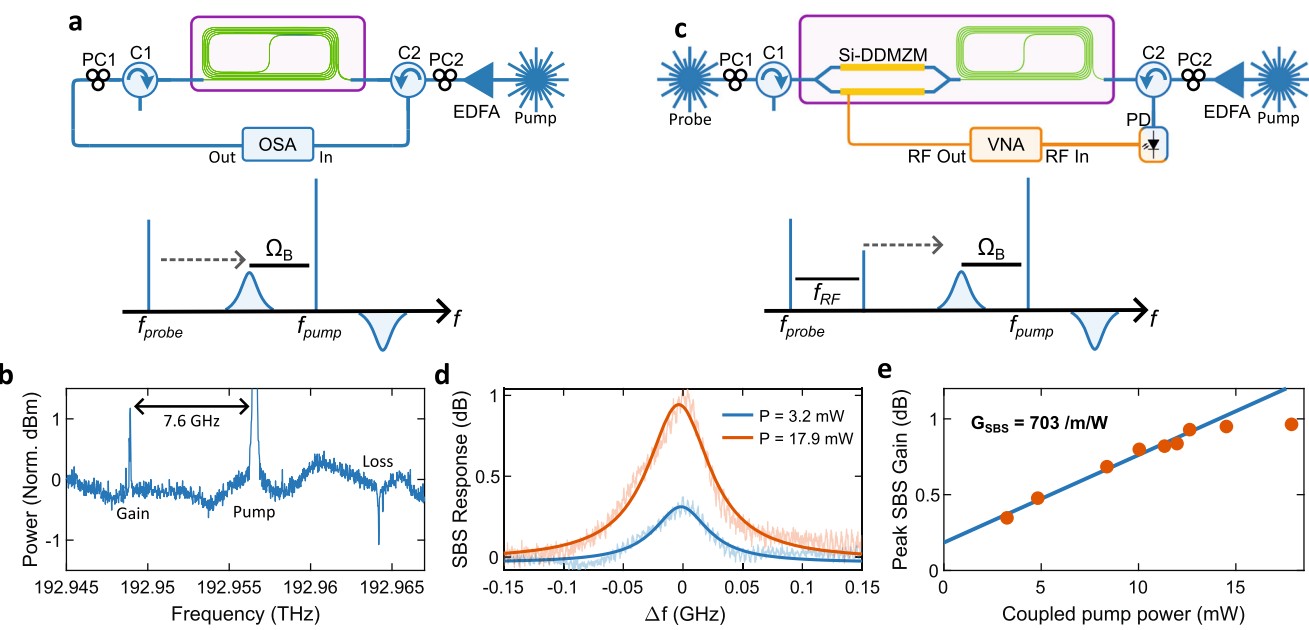

**Fig. 3 | Characterisation of the As₂S₃ Brillouin waveguides. a** Simplified setup for the optical measurement of Brillouin gain. **b** Measured Brillouin gain and loss response. **c** Simplified setup of the electronic measurement of Brillouin gain. **d** The shaded trace represents the measured SBS response in the As₂S₃ waveguides for different coupled pump powers (P). The solid line represents the Lorentzian fit. **e** Measured SBS gain as a function of pump power. OSA Optical Spectrum Analyser (APEX AP2083A), VNA Vector Network Analyser (Agilent PNA N5224A), PD Photo-detector, EDFA Erbium-doped fibre amplifier, DDMZM Dual-drive Mach-Zehnder modulator. The Brillouin frequency shift is indicated by $\Omega_B$.

separated by the Brillouin frequency shift, $\Omega_B$, a strict phase matching condition is met and pump light is reflected off the acoustic wave. The back-scattered pump light is associated with a Stokes gain resonance and is downshifted in frequency, coherently amplifying the probe and resonantly enhancing the stimulated process[47]. An anti-Stokes loss resonance also occurs, which is associated with the annihilation of phonons[51]. A Brillouin frequency shift of 7.6 GHz was measured, as annotated in Fig. 3b. The Brillouin gain and loss resonances are symmetrical around the reflected pump signal.

To characterise the Brillouin gain response of the As₂S₃ wave-guides with more precision, we used an RF vector network analyser (VNA), as shown in the simplified schematic presented in Fig. 3c (see Methods). The measured SBS response for two different coupled pump powers is shown in Fig. 3d. The Lorentzian fit for each trace indicates a peak SBS gain of 0.3 dB (blue) and 1 dB (orange). Figure 3e shows the peak SBS gain as a function of pump power from which we extracted the SBS gain coefficient. SBS gain saturation is observed for peak SBS gain above $\approx 1$ dB, limited by the losses in the As₂S₃

waveguides. Using the measured Si-As$_2$S$_3$ interface loss of 5.5 dB, the fit indicates a Brillouin gain coefficient of 703 /m/W.

## On-chip RF photonic link and custom E-O modulator

We demonstrate the feasibility of the underlying active silicon photonic platform by using the on-chip E-O modulator and PD to construct an on-chip MWP link, as depicted by the schematic in Fig. 4a. We present the measured RF response of the MWP link in Fig. 4b (see Methods). The on-chip link exhibits a 3-dB bandwidth of approximately 15 GHz, with a measured RF link gain of −52 dB at 1 GHz. We see that the RF link gain (blue) is well above the crosstalk that is measured between the RF probes which drive the modulator and PD (orange).

We implemented a custom on-chip E-O modulator that facilitates control of the amplitude and phase of the optical carrier and sidebands. This allows for the implementation of an MWP filter that is switchable between a bandpass and notch response, while also improving the RF link performance[35,52]. Although this modulation scheme has been implemented using a bulk lithium niobate-based dual-parallel Mach-Zehnder modulator in the past[35,52], its photonic integration here shows a greatly reduced footprint and improved bias stability with an on-chip area of only approximately 1 mm × 2 mm.

The schematic of the modulator is depicted in Fig. 4a. CSR control is achieved by embedding an on-chip dual-drive Mach-Zehnder modulator (DDMZM), supplied by the IMEC iSiPP50G PDK[53], in an interferometer. In this demonstration, a 90° RF hybrid coupler was used to generate the RF drive signal in each arm. First, the input optical signal is split into two arms by a tunable coupler (TC1). High-speed E-O modulation occurs in the lower arm of the interferometer using the on-chip DDMZM. The amplitude and phase of the signal in the upper arm of the interferometer are controlled by a variable optical attenuator and tunable phase shifter. When the upper and lower arms of the interferometer are combined by TC2, the resultant optical carrier at the modulator output is the vector sum of the carrier signals in both arms. As a result, the carrier power can be controlled independently of the sideband amplitude and phase, as shown in Fig. 4c. Using the custom on-chip modulator, we control the phase of the resultant optical carrier, which in turn controls the phase of the RF photocurrent contributions, such that they

constructively (blue), or destructively (orange) interfere, as measured using an off-chip PD in Fig. 4d.

## Integrated MHz-resolution MWP notch filter

The integrated Si-As$_2$S$_3$ platform was used to implement an MWP notch filter in the setup depicted in Fig. 5a. A continuous wave laser (L1) is modulated by a VNA, which drives the on-chip DDMZM and forms the Brillouin probe. An RF hybrid coupler imparts a 90° phase shift between the RF drive of each arm of the on-chip DDMZM. The DDMZM is biased, such that the upper and lower sideband magnitudes differ by the amount of applied Brillouin gain and give π out-of-phase photocurrent contributions.

The probe transitions from the Si waveguide to a 5.55 cm long As$_2$S$_3$ waveguide where it undergoes Brillouin processing, which provides narrowband amplification to one sideband, by counter-propagating it against the Brillouin pump, generated by a second laser (L2). Off-chip circulators are used to separate the counter-propagating Brillouin pump and probe. The probe signal subsequently undergoes optical filtering and amplification before photodetection. An RF notch is formed through spectrally-localised destructive interference of RF photocurrents at the PD output, as the magnitude of the Brillouin gain is such that the π out-of-phase photocurrent components are equal in amplitude when the upper sideband coincides with the Brillouin resonant frequency, but unequal otherwise. The on-chip E-O modulator controls the CSR at the output of TC2 to improve the RF link gain (See Supplementary Information Fig. S8–S9).

We did not use the PD on the Si-As$_2$S$_3$ chip for the MWP filtering demonstration because the MWP filter exhibited increased RF losses compared to the intensity-modulation direct-detection (IM-DD) link-only measurement without the As$_2$S$_3$ waveguide. As a result, the MWP link gain of the filter was below the crosstalk between the RF probes that drive the modulator and PD. The main sources of RF loss are attributed to two main factors. The first factor is the increased insertion losses (20 dB) of the As$_2$S$_3$ waveguides, which is linked to the sample preconditioning that removes a 2 μm silica cladding layer to expose the silicon waveguides. The second source of loss is the partial destructive interference of RF photocurrents that are used to generate the MWP notch filter scheme due to the small 0.8 dB of SBS gain

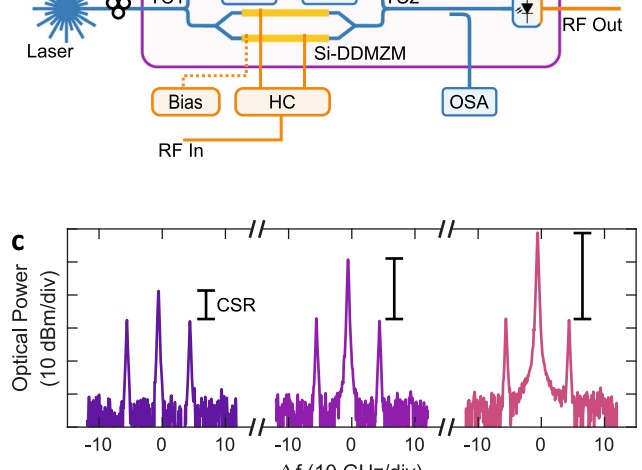

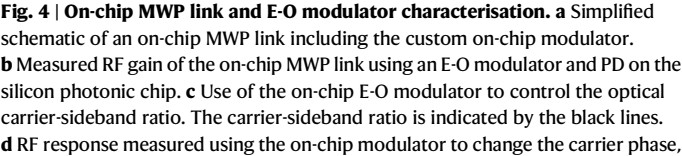

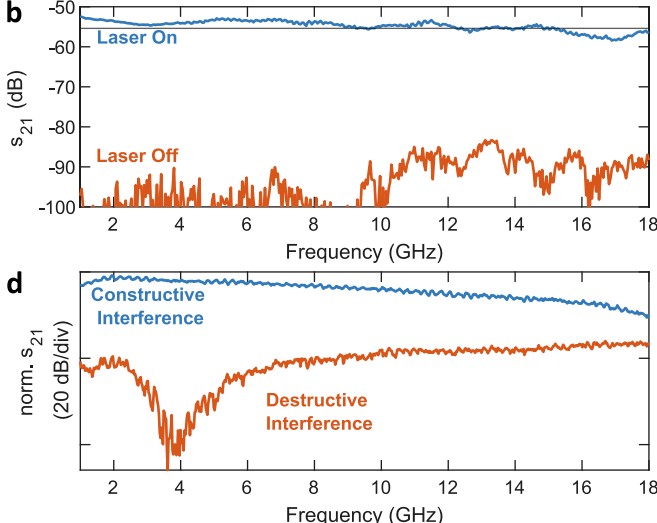

**Fig. 4 | On-chip MWP link and E-O modulator characterisation. a** Simplified schematic of an on-chip MWP link including the custom on-chip modulator. **b** Measured RF gain of the on-chip MWP link using an E-O modulator and PD on the silicon photonic chip. **c** Use of the on-chip E-O modulator to control the optical carrier-sideband ratio. The carrier-sideband ratio is indicated by the black lines. **d** RF response measured using the on-chip modulator to change the carrier phase,

such that the RF photocurrents constructively interfere (blue) or destructively interfere (orange). HC Hybrid coupler, DDMZM Dual-drive Mach-Zehnder modulator, PD Photodetector, TC Tunable coupler, TPS Tunable phase shifter, VOA Variable optical attenuator, OSA Optical spectrum analyser (II–VI WaveAnalyser), CSR Carrier-sideband ratio.

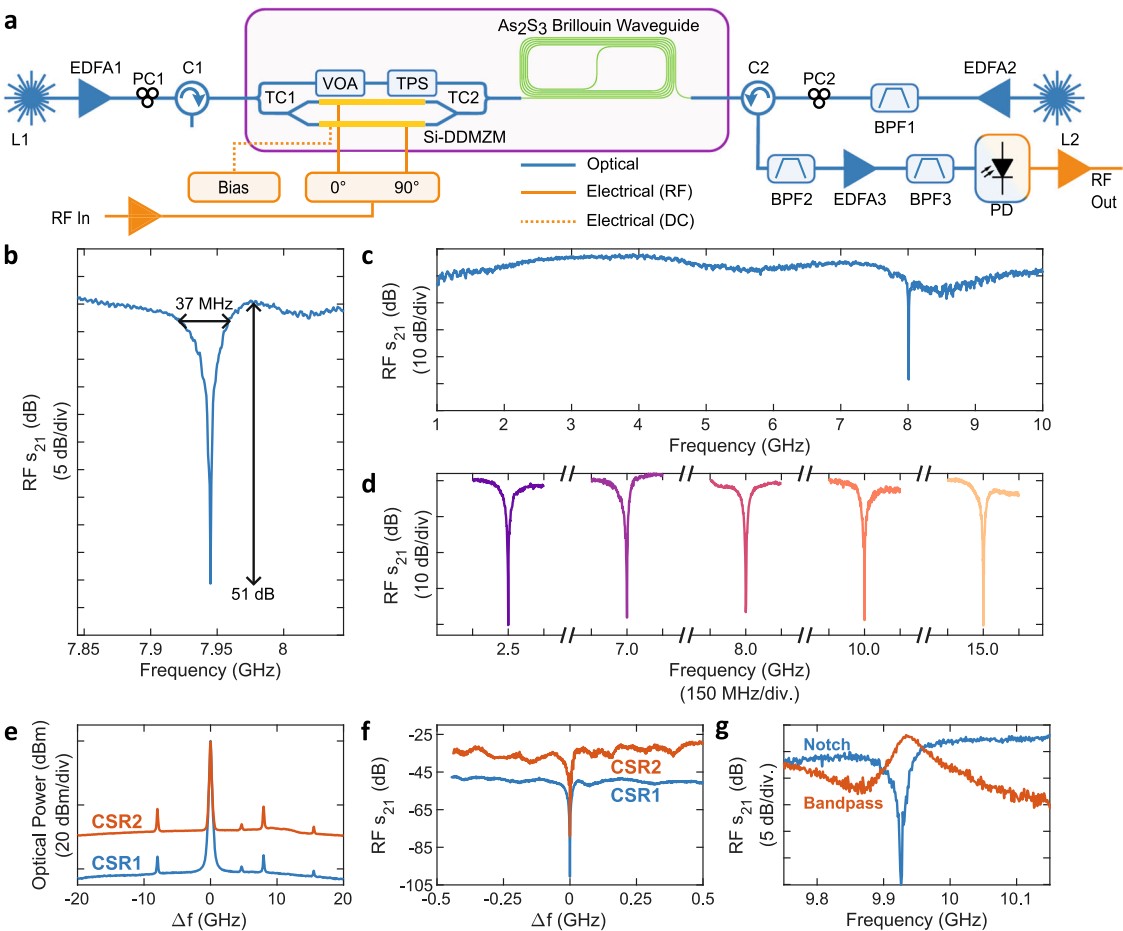

**Fig. 5 | Si-As$_2$S$_3$ MWP notch filter. a** Experimental setup. **b** Measured RF notch filter response with the narrow 3-dB bandwdith of 37 MHz and large 51 dB rejection annotated. **c** Measured RF response of the MWP notch filter with a notch at 8 GHz. **d** Notch filter frequency tuning. The notch frequency is changed between 2.5–15 GHz. **e** Measured optical spectra for two different CSRs. Optical processing on the Si-As$_2$S$_3$ PIC reduces the CSR from CSR1 (blue) to CSR2 (orange). The modulated sideband is offset by 7 GHz from the optical carrier. **f** Measured RF response for notch filters with CSR1 and CSR2. **g** Measured notch and bandpass filter responses. A bandpass response can be created by changing the DC bias of the on-chip DDMZM, such that the photocurrents destructively interfere outside the Brillouin resonance. L Laser (Teraxion NLL), EDFA Erbium-doped fibre amplifier, PC Polarisation controller, C Circulator, TC Tunable coupler, VOA Variable optical attenuator, TPS Tunable phase shifter, DDMZM Dual-drive Mach-Zehnder modulator, BPF Band-pass filter (BPF1, BPF3: Santec OTF-320. BPF2: Alnair BVF-300CL).

generated on the Si-As$_2$S$_3$ chip (See discussion in Supplementary Information section IIIC and Fig. S10–S11). These RF losses can feasibly be overcome by improving the As$_2$S$_3$ post-processing fabrication flow, reducing sidewall scattering through further design optimisation of As$_2$S$_3$ waveguide dimensions, and avoiding the silica cladding etching step by including dedicated open sections on the initial silicon circuit. Reducing the As$_2$S$_3$ waveguide losses will improve the link gain through two mechanisms, a direct improvement from lower optical insertion loss and larger Brillouin gain via the increase in effective length.

Figure 5b presents an RF notch response with narrow 37 MHz 3-dB bandwidth and deep 51 dB RF rejection, generated using only 0.8 dB SBS gain. The broadband response of the MWP filter, with a notch at 8 GHz, is shown in Fig. 5c. The ripple in the passband is caused by reflections at the Si-As$_2$S$_3$ interface (See discussion in Supplementary Information section I and Fig. S2). Figure 5d shows notch frequency tunability between 2.5 and 15 GHz. The notch frequency was tuned by changing the frequency of the Brillouin pump relative to the probe. The maximum notch frequency is currently limited by the limited magnitude of generated SBS gain and the roll-off of the on-chip E-O modulator response. Additionally, the RF notch depth is sensitive to amplitude and phase perturbations of the optical carrier and sidebands. Such perturbations can arise from

changes to the DDMZM bias voltage (See Supplementary Information Fig. S12). However, with reduced As$_2$S$_3$ losses, the magnitude of applied Brillouin gain will increase, enabling a larger notch frequency tuning range and reducing the sensitivity of the RF notch depth to environmental factors.

We demonstrate CSR control using the on-chip modulator in Fig. 5e. We present the measured optical spectrum for two different CSRs, using EDFA3 to maintain constant power at the PD. The corresponding RF response for CSR1 and CSR2 is shown in Fig. 5f. Using the on-chip modulator, we demonstrate a 13 dB improvement to the RF link gain through CSR control. The weak seed power into the EDFA causes significant amplified spontaneous emission (ASE) noise in the EDFA output. The excessive ASE noise is expected to be reduced with lower As$_2$S$_3$ losses, such that a net RF link gain and NF improvement is observed with CSR suppression, as observed in ref. 54.

The flexibility of MWP signal processing is highlighted through our use of the on-chip E-O modulator to alternate between bandpass and notch responses in Fig. 5g. We reconfigure the filter response between notch (blue) and bandpass (orange) filter responses by changing the modulator bias. For a bandpass response, we set the sideband amplitudes to be equal outside the Brillouin resonance, such that the photocurrents destructively interfere. At present, the rejection ratio of the bandpass filter response is limited due to partial

broadband destructive interference caused by residual dispersion of the on-chip components.

## Discussion

In this section, we outline improvements that can be made to the Si-As$_2$S$_3$ platform regarding optical loss and RF link performance. Reducing the As$_2$S$_3$ losses greatly improves RF link performance through two distinct mechanisms, the reduced losses of the As$_2$S$_3$ waveguides themselves and from the resulting increase in generated Brillouin gain. The first improvement to the link performance is due to the increased optical power at the PD from the reduced insertion losses. The optical power at the PD would increase by 12.8 dB if the As$_2$S$_3$ waveguides on the Si-As$_2$S$_3$ platform exhibited the same 0.4 dB/cm propagation loss as measured on the standalone As$_2$S$_3$ platform and the same 0.1 dB Si-As$_2$S$_3$ interface loss reported by ref. 49. The second source of RF link performance improvement is from an increase in Brillouin gain, owing to the increased effective length of the As$_2$S$_3$ waveguide. This reduces the level of destructive interference in the filter passband. With 0.4 dB/cm propagation loss, the effective length of the 5.55 cm As$_2$S$_3$ waveguide would increase from 1.98 to 4.24 cm. As a result, we can feasibly expect the SBS gain to increase from 0.8 dB to 18 dB (See discussion in Supplementary Information section IIB and Table S1). The combined effect of reduced As$_2$S$_3$ losses leads to a significant RF link performance improvement compared to the results that we presented here of 46 dB to the link gain, as well as a NF reduction and SFDR$_3$ increase by 20 dB, respectively (See discussion in Supplementary Information section IIIC1). Furthermore, recent progress of integrated photonic circuits has led to the rapid development of low-$V_{\pi_{RF}}$ modulators that would lead to direct RF link performance improvements[55,56].

We acknowledge the link performance that we demonstrate here does not yet meet the requirements for real-world systems. However, significant RF link improvements outlined above indicate that the MWP link gain can be significantly improved, providing the opportunity to greatly reduce system SWaP by simultaneously using integrated E-O modulators, PDs and Brillouin waveguides. Additionally, the observed MWP filter 3-dB bandwidth of 37 MHz can be further reduced with lower As$_2$S$_3$ waveguide losses, which increases the magnitude of SBS gain and causes an associated gain narrowing[57], with a 3-dB linewidth of 10 MHz having been measured from 52 dB on-chip gain[46]. Further bandwidth reduction can be achieved by exploiting fundamental properties of the Brillouin interaction such as increasing photon lifetime through dynamic gratings[58], or tailoring the response of overlapping Brillouin gain and loss resonances[59]. Similarly, incorporating intermodal SBS[24], forwards SBS[60], or using tunable couplers to separate the counter-propagating pump and probe could enable systems where non-reciprocal devices are not needed, simplifying the overall system design.

Table 1 summarises the performance of state-of-the-art MWP filters which exhibit a high degree of integration. It shows that

this work is key in unifying the advantages of nonlinear Brillouin signal processing, which offers fine spectral resolution and reconfigurability, with active E-O components. An additional benefit of our work is the flexibility enabled by the on-chip E-O modulator, enabling RF link gain improvement through CSR control and simultaneously facilitates the MWP filter to be altered between bandpass and notch responses.

The advantage of this work is further highlighted by comparing the reported 3-dB bandwidth of the MWP filters. Although some MRR and tapped delay line demonstrations have successfully integrated on-chip E-O devices[10,20], they were generally limited to 3-dB bandwidths that exceeded hundreds of MHz. Additionally, Brillouin processing offers advantages over alternate optical signal processing methods. For example, the FSR of MRRs and tapped delay line MWP filters limit their frequency tunability. However, Brillouin MWP filters do not face such limitations as there is no FSR associated with the SBS process. Furthermore, the thermal stability of the Brillouin frequency shift (≈1 MHz/°C in optical fibres[25]) ensures stable notch operation, whereas the frequency of high-Q MRRs often exhibits a strong dependency on temperature (e.g., ≈12 GHz/°C was reported for a silicon MRR with Q of 10$^7$ in ref. 16), which introduces a challenging requirement to either stabilise the MRR, or lock the laser to the MRR resonant frequency. Although the 21 MHz 3-dB bandwidth demonstrated by the MRR-based MWP filter in ref. 16 is promising for future generations of MRR-based systems, the device did not include an E-O interface and faces challenges from thermal sensitivity and the presence of an FSR in its response, as outlined above.

We have presented an integrated MWP notch filter based on heterogeneously integrated As$_2$S$_3$ waveguides in an active silicon photonic platform. This platform enabled an MWP filter with fine spectral resolution of 37 MHz and large RF rejection of 51 dB. Our demonstration addressed the challenge of achieving fine spectral resolution MWP processing and system integration in a foundry-manufactured silicon photonic platform. Furthermore, we improved the RF link performance of the filter through CSR control using a custom on-chip E-O modulator. With future optimisation of this device and integration of a laser, fully integrated Brillouin-based MWP systems can be feasibly achieved. This enables a new class of integrated platforms to implement a range of on-chip MWP processing functionalities, including phase shifters, true time delays, frequency converters and RF sources. Our work represents a significant step toward the ultimate goal of achieving a high-performance fully-integrated MWP processor with fine spectral resolution and broadband tunability.

## Methods

### Brillouin response measurements

We measured the SBS response using both optical and electronic techniques. For the optical measurement, we used the tracking

**Table 1 | Summary of state-of-the-art integrated MWP filters**

| Year | Material Platform | Filter Type | Optical Processing | On-chip E-O? | FSR-free operation? | On-chip MWP link improvement? | 3-dB Bandwidth (MHz) | Ref |
|------|-------------------|-------------|--------------------|--------------|---------------------|-------------------------------|----------------------|-----|
| 2021 | Si + InP | Notch, Bandpass | MRR | **Y** | N | N | 360–470 | 10 |
| 2022 | Si + InP + GaAs | Bandpass | TDL | **Y** | N | N | 900–1100 | 20 |
| 2022 | Si$_3$N$_4$ | Notch, Bandpass | MRR | N | N | **Y** | 400 | 11 |
| 2022 | Si | Bandpass | MRR | N | N | N | 20.6 | 16 |
| 2020 | As$_2$S$_3$ | Bandpass | SBS | N | **Y** | N | 30–350 | 35 |
| 2022 | Si | Notch | FSBS | N | **Y** | N | 2.7 | 36 |
| 2023 | Si + As$_2$S$_3$ | Notch, Bandpass | SBS | **Y** | **Y** | **Y** | 37 | This work |

*MRR* microring resonator, *TDL* tapped delay line, *SBS* Stimulated Brillouin scattering, *FSBS* forward SBS.
*E-O* Electro-optic. Bold text is used to indicate the desired characteristic.

generator function of an optical spectrum analyser (OSA) to measure the Brillouin response of the $As_2S_3$ waveguides. The output of the OSA, which constitutes the Brillouin probe, was counter-propagated against the Brillouin pump in the $As_2S_3$ waveguide and was swept through the Brillouin gain and loss resonances. Off-chip circulators were used to separate the pump and probe, and polarisation controllers were used to ensure the pump and probe were in the $TE_{00}$ polarisation, verified by maximising power transmission through the Si-$As_2S_3$ chip.

For the electrical measurement, we used an RF VNA to measure the Brillouin gain response with greater resolution and spectral precision. We drove the on-chip DDMZM with the output of the VNA. The DDMZM was biased to give single-sideband modulation, ensuring the measured RF transfer function was proportional to the optical transfer function between the modulator and PD. The modulated sideband was swept through the Brillouin gain resonance. We recorded the RF response of the MWP link for multiple pump powers and divided the pump-on response by the pump-off response to determine the SBS gain. A linear slope was then subtracted from the measured response to compensate for roll-off and a Lorentzian curve was fitted to extract the peak Brillouin gain.

### On-chip MWP link measurement

We use a laser around 1550 nm with an on-chip power of only −8.6 dBm. We used RF probes to drive the RF arms of the on-chip DDMZM, using a 180° RF hybrid coupler to induce a phase shift between each arm. The on-chip tunable couplers TC1 and TC2 route the signal to the on-chip DDMZM and bypass upper arm of the larger interferometer. We measure the RF response using a VNA and subtract the measured 0.48 dB/GHz roll-off of the driving RF electronics from the MWP link response. The measurement does not include RF amplification.

### Data availability

The experimental data presented in this study is available at: https://doi.org/10.5281/zenodo.8411989.

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

## Acknowledgements

The authors acknowledge the facilities as well as the scientific and technical assistance of the Research and Prototype Foundry Core Research Facility at the University of Sydney, part of the NSW node of the NCRIS-enabled Australian National Fabrication Facility. This work has been made possible through the access to the ACT node of the Australian National Fabrication Facility (ANFF-ACT). We acknowledge the support Australian Research Council (DP220101431 and DP200101893).

## Author contributions

Y.L., A.C.B. and B.J.E. conceived the high-level concept. Y.L, M.G. and A.C.B. performed the architectural design of the photonic integrated circuit. M.G., Y.L. and A.C.B. performed the design and layout of the photonic integrated circuit. C.T.B., D.Y.C. and S.J.M. performed the post-processing on the photonic chip to integrate the As$_2$S$_3$ waveguides. M.G. performed the As$_2$S$_3$ waveguide simulations with input from C.K.L. M.G. performed the experiments with input from M.M., Y.L. and A.C.B. M.G., Y.L., M.M. and B.J.E. wrote the manuscript with contributions from all authors. A.C.B led and supervised the heterogeneous integration. B.J.E. supervised the project.

## Competing interests

The authors declare no competing interests.
