## [Peer Review File · Nature Communications]

REVIEWER COMMENTS

Reviewer #1 (Remarks to the Author):

The paper by Matthew Garrett et al. presents an Si-As₂S₃ heterogeneous integrated microwave photonic notch filter, utilizing the Brillouin effect to achieve a narrow bandwidth of 37 MHz. While the authors have previously proposed the concept of Si-As₂S₃ heterogeneous integration, the novelty of this work lies in the integration of nonlinear functionality and electro-optic (E-O) conversion, enabled by the chalcogenide waveguide and active silicon photonic devices, such as the high-speed modulator. Additionally, the authors demonstrate an on-chip version of the CSR control modulator, showcasing a significant 13 dB improvement in the RF link gain. The manuscript has the potential to be published if the following concerns or suggestions are adequately addressed:

1. Page 5, Section B, Paragraph 1: The sentence "the RF link gain (blue) is well above the crosstalk..., which inhibits the use of the on-chip PD for the MWP filtering demonstration" is unclear. Why the on-chip PD is inhibited if the results is well above the crosstalk? It seems the on-chip PD are not used in the characterization of the notch filter. Is that due to the on-chip loss and limited responsivity? The author should comment this in the main text.
2. It would be valuable to discuss any potential methods for further narrowing down the bandwidth in the As₂S₃ platform, considering that a ~3.5 MHz Brillouin filter was achieved in [Gertler, Shai, et al., APL Photonics 5.9 (2020): 096103]. Furthermore, a comparison with state-of-the-art microring-based microwave photonic filters, which typically exhibit similar 3-dB bandwidths (~20 MHz) [Zhang, Long, et al., Laser & Photonics Reviews 16.4 (2022): 2100292], could provide insightful analysis of the SBS filtering structure.
3. The demonstrated filter exhibits a 51 dB rejection ratio at 8 GHz. It would be beneficial for the authors to address the sensitivity of the filter to system configurations, such as the bias voltage of the DDMZM and the pumping wavelength, considering the limited SBS gain of approximately 1 dB due to waveguide losses. Additionally, showcasing a wider frequency range in Figure 5(d) would enhance the presentation. Unlike the resonance based filter, SBS scheme is FSR-free, that is also one of the advantages that should be emphasized.
4. While the characterizations of the microwave photonic filter are based on small signal testing using a VNA, it would be advantageous to include results that demonstrate the suppression of real-world microwave signals, similar to the approach in [Tao, Yuansheng, et al., Photonics Research 9.8 (2021): 1569-1580], for verification.

5. In the main text, it is important to provide a detailed discussion of the fabrication processes employed to reduce waveguide losses.

6. Subscript should be used in the title of Fig.1 b

7. Since bi-directional pumping necessitates the use of a circulator, which poses challenges for chip-based integration, it would be valuable to clarify whether this Si-As₂S₃ platform supports forward SBS as an alternative approach.

8. The utilization of the novel modulation control scheme results in a higher carrier-to-signal ratio. However, the extra link gain is actually provided by the external optical amplifier. It is crucial to consider the influence of additional amplified spontaneous emission (ASE) noise introduced by the external optical amplifier on the filtering performance. The authors should discuss the impact of this noise.

In all, I find this work, although constrained by losses, is still an interesting attempt to integrate the nonlinear functions into the foundry-based active silicon photonic circuits.

Reviewer #2 (Remarks to the Author):

In this research article, the authors propose a compact 5mmx 5mm chip-scale microwave photonic notch filter include active E-O components and heterogeneously integrated As₂S₃ waveguides, which can provide a Brillouin gain of 44.5dB to the link. The reported filter achieves a fine spectral resolution and RF rejection of 37MHz and 51dB, respectively. In addition, the proposed filter demonstrates a potential of heterogeneous integration based on silicon photonic platform and Brillouin-based waveguides in implementing on-chip MWP processing functionalities, including phase shifters, true time delays, frequency converters and RF sources. Also, the paper and supplementary material are well written and ideas are well explained with adequate simulations and experimental results to support the claims. Here are my comments and question for the authors:

1. I understand that the Brillouin gain is highly desired to MWP filter, and with the heterogeneous integrations of Brillouin Si-As₂S₃ waveguides, it is possible to achieve a fully integrated MWP system, however, what is the price of implementing such heterogeneous integrations?

2. In order to show the advantage of proposed heterogeneous integration, it is better to give a performance summary or comparison between this work and other reported state-of-art silicon photonic microwave filters.
3. How much does the performance of Brillouin Si-As₂S₃ waveguides change with the temperature? Or is there any unique requirement to silicon photonic process to achieve the proposed heterogeneous integration?
4. The lengths of Si-As₂S₃ in supplementary materials (Table S1) and the manuscript (session III) are different, please check the value.
5. Is the photodetector in Fig.1 only used for the characterization of E-O modulator? Why do not used it to replace the off-chip PD in Fig.5 to increase the integration density of Si-As₂S₃ MWP notch filter. Is there any other design concern?
6. Is the proposed Si-As₂S₃ waveguide sensitive to the polarization?

Please note that throughout this document, comments from the reviewers are shown in black, our responses to the reviewers are included in blue following each of the reviewers' comments and text added to the updated manuscript, is shown in red.

Reviewer #1 (Remarks to the Author):

The paper by Matthew Garrett et al. presents an Si-As₂S₃ heterogeneous integrated microwave photonic notch filter, utilizing the Brillouin effect to achieve a narrow bandwidth of 37 MHz. While the authors have previously proposed the concept of Si-As₂S₃ heterogeneous integration, the novelty of this work lies in the integration of nonlinear functionality and electro-optic (E-O) conversion, enabled by the chalcogenide waveguide and active silicon photonic devices, such as the high-speed modulator. Additionally, the authors demonstrate an on-chip version of the CSR control modulator, showcasing a significant 13 dB improvement in the RF link gain. The manuscript has the potential to be published if the following concerns or suggestions are adequately addressed:

We thank the reviewer for the positive feedback and for highlighting the novelty of this work in integrating nonlinear processing and active silicon photonic devices on a single chip for a compact MWP processing device.

1. Page 5, Section B, Paragraph 1: The sentence "the RF link gain (blue) is well above the crosstalk..., which inhibits the use of the on-chip PD for the MWP filtering demonstration" is unclear. Why the on-chip PD is inhibited if the results is well above the crosstalk? It seems the on-chip PD are not used in the characterization of the notch filter. Is that due to the on-chip loss and limited responsivity? The author should comment this in the main text.

We thank the reviewer for their comments which helped to improve the clarity of the manuscript. The measured RF link gain was above the crosstalk when we used an intensity-modulation direct-detection (IM-DD) MWP link with the on-chip high-speed modulators and PDs (1 A/W responsivity), while bypassing the As₂S₃ waveguides. However, with the As₂S₃ waveguide between the on-chip modulator and the PD, the insertion loss of the As₂S₃ waveguide reduced the link gain below the level of the crosstalk between the RF probes that drive the modulator and the PD. The additional 12.3 dB of optical insertion loss from the As₂S₃ waveguides increases the RF link loss by 24.6 dB, according to the square-law photodetection. Additionally, when performing the filter functionality, the strong destructive RF interference (due to the small SBS gain in current devices) contributed another 20 dB of RF loss. These losses lead to an RF link gain below the crosstalk. Therefore, the on-chip PD was not used for the MWP filtering demonstration.

Here, we envisage two possible future solutions. Firstly, in the future, with reduction in the As₂S₃ waveguide losses the RF link gain of the MWP filter will no longer be below the crosstalk between the probes and we expect that the on-chip PD can be used for the MWP filtering demonstration. Secondly, with on-chip amplification using semiconductor optical

amplifiers [7], [8] or rare-earth-ion-doped waveguide amplifiers [9], the optical insertion loss can be compensated, and thus the on-chip PD can be deployed for a fully-integrated microwave photonic filter systems.

Further discussion of the main sources of loss in the MWP filtering demonstration is included below. A more comprehensive discussion can be found in Section III C of the Supplementary Information.

- **On-chip losses of the As₂S₃ waveguides:** The propagation losses and interfaces losses of the As₂S₃ waveguides in the Si-As₂S₃ platform (an average of 1.7 dB/cm and 6.3 dB, respectively) are much higher than the previously reported by Morrison et. al in a passive Si-As₂S₃ platform (0.7 dB/cm and 0.1 dB), or as we measured on a standalone As₂S₃ chip (0.4 dB/cm). This was deduced to be due to the initial wet etching process required for exposing the Silicon waveguides. The increased As₂S₃ losses on the Si-As₂S₃ platform introduced an additional 12.3 dB of optical loss. Due to the square law nature of photodetection, this corresponds to an RF loss of 24.6 dB.
- **Interferometric optical modulation scheme:** The on-chip MWP link characterisation presented used a modulation with two in-phase RF photocurrent contributions. However, the output photocurrents of the MWP notch filter consist of two photocurrent contributions that are π out-of-phase. This has the overall effect of reducing the RF link gain of the MWP filter compared to the on-chip MWP link characterisation. Because a modest 0.8 dB of Brillouin gain was used in this work, the photocurrent contributions are almost equal in magnitude and therefore almost cancel. With reduced As₂S₃ losses, the Brillouin gain will increase, and therefore increase the resultant RF photocurrent power. If As₂S₃ losses of 0.7 dB/cm can be achieved, we expect to generate 17.8 dB of SBS gain for the same pump power. The corresponding RF link gain improvement from the reduced level of photocurrent cancellation would be 19.9 dB. Therefore, a total RF link gain improvement of 24.6 + 19.9 = 44.5 dB is to be expected, which brings the link gain above the crosstalk and enable the use of the on-chip PD in the MWP filter demonstration.

To improve the clarity of the revised manuscript, we have:

- (1) Removed the statement that the reviewer pointed out was unclear '*which inhibits the use of the on-chip PD for the MWP filtering demonstration.*' The new passage in Section II B reads:

"We see that the RF link gain (blue) is well above the crosstalk that is measured between the RF probes which drive the modulator and PD (orange)."

- (2) Added the following discussion as to why we did not use the on-chip PD for the MWP filtering demonstration directly after we introduce the MWP filter results in Section II C of the updated manuscript:

“We did not use the PD on the Si-As₂S₃ chip for the MWP filtering demonstration because the MWP filter exhibited increased RF losses compared to the intensity-modulation direct-detection (IM-DD) link-only measurement without the As₂S₃ waveguide. As a result, the MWP link gain of the filter was below the crosstalk between the RF probes that drive the modulator and PD. The main sources of RF loss are attributed to two main factors. The first factor is the increased insertion losses (12.3 dB) of the As₂S₃ waveguides, which is linked to the sample preconditioning that removes a 2 μ m silica cladding layer to expose the silicon waveguides. The second source of loss is the partial destructive interference of RF photocurrents that are used to generate the MWP notch filter scheme due to the small 0.8 dB of SBS gain generated on the Si-As₂S₃ chip (See Supplementary Information Section III C). These RF losses can feasibly be overcome by improving the As₂S₃ post-processing fabrication flow, reducing sidewall scattering through further design optimisation of As₂S₃ waveguide dimensions, and avoiding the silica cladding etching step by including dedicated open sections on the initial silicon circuit. Reducing the As₂S₃ waveguide losses will improve the link gain through two mechanisms, a direct improvement from lower optical insertion loss and larger Brillouin gain via the increase in effective length.”

2. It would be valuable to discuss any potential methods for further narrowing down the bandwidth in the As₂S₃ platform, considering that a ~3.5 MHz Brillouin filter was achieved in [Gertler, Shai, et al., APL Photonics 5.9 (2020): 096103]. Furthermore, a comparison with state-of-the-art microring-based microwave photonic filters, which typically exhibit similar 3-dB bandwidths (~20 MHz) [Zhang, Long, et al., Laser & Photonics Reviews 16.4 (2022): 2100292], could provide insightful analysis of the SBS filtering structure.

We acknowledge the suggestion of discussing possible approaches to reducing the filter bandwidth. As pointed out by the reviewer Brillouin-based schemes have the potential for even higher resolution, as demonstrated in the forward direction for coupled waveguides [6], [10] and tapers [11], and in the backward direction with dynamic gratings [12]. Additionally, pump spectrum tailoring [13] has enabled narrow linewidths of 3.4 MHz. However, these methods come at the cost of increased complexity.

The most feasible way to reduce the 3-dB bandwidth of Brillouin-based filter is to increase the magnitude of the SBS gain generated in the As₂S₃ waveguides by reducing the insertion loss. This is because lower As₂S₃ waveguide losses lead to an increased magnitude of generated Brillouin gain, and the linewidth of the Brillouin resonance reduces with increasing magnitudes of SBS gain [14]. For example, a linewidth of 10 MHz was measured from 52 dB of on-chip SBS gain [15]. Furthermore, the RF notch is formed through the addition of two photocurrents. This interferometric process further reduces the 3-dB bandwidth of the RF notch with increasing SBS gain [16].

Regarding MWP filters that are based on MRRs, we acknowledge that some state-of-the-art MRRs are starting to exhibit 3-dB bandwidths on the order of tens of MHz. However,

there are specific advantages of using nonlinear Brillouin processing over MRRs. Firstly, the Brillouin gain response does not exhibit a free-spectral range (FSR) and hence has no limitations in the operating frequency range of the MWP processor.

Secondly, MRRs typically exhibit relatively poor thermal stability to the extent that if a high-powered signal is coupled into a ring, it can cause self-heating, due to increased intra-cavity power, and detune the resonant frequency of the ring. This increases system complexity in locking the laser to the MRR resonance. For MRR-based MWP notch filters, this would result in a shift of the notch frequency as the RF power increases, thereby not filtering the desired frequency and limiting the dynamic range of the underlying MWP system. As an example, the MRR presented in [4] exhibits a thermal drift of approximately 12 GHz/°C. In contrast, the relatively small temperature dependence of the Brillouin frequency shift (BFS) has been measured at 1.0580 MHz/°C in polarisation maintaining silica fibres [17]. The increased frequency stability of Brillouin based processing is highly advantageous for MWP processing applications.

Furthermore, the SBS gain response is highly reconfigurable, as the Brillouin gain spectrum can be controlled by shaping the spectrum of the optical pump used to generate the Brillouin interaction [18], [19]. Although a reconfigurable response can be achieved using MRR-based processing, it is difficult to achieve as it requires a bank of MRRs to be individually controlled. The thermal sensitivity of MRRs, outlined above, and thermal crosstalk present significant challenges for MRRs to achieve the same frequency reconfigurability as a Brillouin-based approach.

We have included a discussion on the potential to further reduce the Brillouin resonant linewidth in the updated manuscript in Section III:

“...Additionally, the observed MWP filter 3-dB bandwidth of 37 MHz can be further reduced with lower As_2S_3 waveguide losses, which increases the magnitude of SBS gain and causes an associated gain narrowing [57], with a 3-dB linewidth of 10 MHz having been measured from 52 dB on-chip gain [46]. Further bandwidth reduction can be achieved by exploiting fundamental properties of the Brillouin interaction by increasing photon lifetime through dynamic gratings [58], or tailoring overlapping Brillouin gain and loss resonances [59]...”

We added to the discussion of MRR-based MWP filters in Section I of the updated manuscript:

“Although ultra-high-quality factor (Q) MRRs can be fabricated, they are extremely sensitive to input signal power and temperature fluctuations due to thermo-optic effects [15, 16], which can add challenges in aligning the laser to the MRR resonance. Additionally, the free-spectral range (FSR) of the MRR can limit the frequency operating range of the resultant MWP processor.”

“Similar to MRR-based processors, the periodic response of tapped delay line MWP processors can limit their frequency operating range.”

“In addition to fine spectral resolution, SBS offers several advantages for MWP processing applications as the Brillouin response does not exhibit an FSR which, does not impose any limitations on the operating range of the resultant MWP system [1, 2]. Furthermore, the thermal stability of the Brillouin frequency shift (BFS), ≈ 1 MHz/°C in optical fibres [25], can be much greater than the thermal stability of MRRs, for example a silicon MRR with Q of 10^7 exhibited a thermal sensitivity of ≈ 12 GHz/°C [16]. This highlights the advantages of SBS for MWP processing in terms of tunability and stability.”

3. The demonstrated filter exhibits a 51 dB rejection ratio at 8 GHz. It would be beneficial for the authors to address the sensitivity of the filter to system configurations, such as the bias voltage of the DDMZM and the pumping wavelength, considering the limited SBS gain of approximately 1 dB due to waveguide losses. Additionally, showcasing a wider frequency range in Figure 5(d) would enhance the presentation. Unlike the resonance-based filter, SBS scheme is FSR-free, that is also one of the advantages that should be emphasized.

We thank the reviewer for the insightful comments and again highlighting the advantage of our scheme (FSR free). We briefly summarise our response and the changes we made to the manuscript here before providing the detailed text added to the manuscript below.

- From the calculations we present below, a bias drift of less than 4 mV will still maintain large filter extinction ratio exceeding 40 dB. The typical noise of a DC power supply (Rigol DP832) is <350 μ Vrms and long-term stability is approximately 2 mV. This indicates that large rejection can be achieved over a long time periods using off the shelf electronic components. Additionally, the slowly varying thermal drift of the modulator bias can be compensated for by using feedback systems that are widely used in data communications applications.
- Changing the pumping wavelength will change the frequency of the notch in the RF domain by the same amount but has minimal effect on the notch depth.
- Increasing the magnitude of applied Brillouin gain improves the stability of the notch depth and has a positive effect on the link gain.

We added the following passages in Section II C of the updated manuscript to reflect the above discussion:

“...Additionally, the RF notch depth is sensitive to amplitude and phase perturbations of the optical carrier and sidebands. Such perturbations can arise from changes to the DDMZM bias voltage (See Supplementary Information Section III C). However, with reduced As_2S_3 losses, the magnitude of applied Brillouin gain will increase, enabling a larger notch frequency tuning range and reducing the sensitivity of the RF notch depth to environmental factors.”

“The notch frequency was tuned by changing the frequency of the Brillouin pump relative to the probe.”

With regards to the advantages of SBS-based processing being FSR-free, we have updated the Section I of the manuscript to highlight this advantage in the following passage (this was covered in question two from the same reviewer):

“In addition to fine spectral resolution, SBS offers several advantages for MWP processing applications as the Brillouin response does not exhibit a FSR, which does not impose any limitation on the operating range of the resultant MWP system [1, 2]...”

Detailed analysis was added to the manuscript and supplementary material based on the reviewers suggestion. Please see the revisions to the manuscript below.

DDMZM Bias Voltage:

Note, we have added the following discussion to Section III C 3 of the Supplementary material of the updated manuscript.

For a notch to form, the resulting photocurrent contributions from each sideband must be equal in amplitude and π out-of-phase. The DDMZM bias voltage changes the amplitude of each sideband. Therefore, changing the DDMZM bias voltage affects the RF notch depth.

To analyse the dependency of the RF notch depth on DDMZM bias voltage, let us consider the spectral components of the DDMZM output under small-signal modulation. These equations are given in eq.'s S7-S9 in the supplementary information and repeated in below in eq.'s (R1)-(R3) for convenience:

$$E_{-1} = \frac{E_0}{2} J_1(\beta) (je^{j\phi_B} + 1) e^{j(\omega_c - \omega_{RF})t} \quad (R1)$$

$$E_{C1} = \frac{E_0}{2} J_0(\beta) (e^{j\phi_B} + 1) e^{j\omega_c t} \quad (R2)$$

$$E_{+1} = \frac{E_0}{2} J_1(\beta) (je^{j\phi_B} - 1) e^{j(\omega_c + \omega_{RF})t} \quad (R3)$$

Where E_0 is the electric field of the laser at the input of the DDMZM, J_n is the nth order Bessel function of the first kind, β is the modulation index defined as $\beta = \frac{\pi V_{RF}}{V_{\pi,RF}}$ where V_{RF} is the RF driving voltage of the DDMZM, $V_{\pi,RF}$ is the half-wave voltage of the DDMZM, ϕ_B is the DC phase angle of the DDMZM, and ω_c and ω_{RF} are the angular frequencies of the laser driving RF signals, respectively.

In the filter scheme presented in this work, the upper sideband is amplified by SBS gain to form the filter notch. The upper sideband after SBS amplification by a factor of G can be expressed as:

$$E_{+1} = \frac{E_0}{2} \sqrt{G} J_1(\beta) (je^{j\phi_B} - 1) e^{j(\omega_c + \omega_{RF})t} \quad (R4)$$

The RF output is obtained through photodetection. The RF photocurrent is related to the optical fields through the expression:

$$i(\omega_{RF}) \propto |E_{C1}||E_{+1}|e^{j(\phi_{+1}-\phi_{C1})} + |E_{C1}||E_{-1}|e^{j(\phi_{C1}-\phi_{-1})} \quad (R5)$$

In this work, the DDMZM bias was set such that the upper and lower sideband photocurrent contributions were π out of phase and differed in power by the magnitude of applied SBS gain. Assuming 1 dB of applied SBS gain, a DC bias angle of $\phi_B = -0.11$ radians is required. The DDMZM used in this work has a DC half-wave voltage of ≈ 12 V, which corresponds to a voltage of 0.44 V.

Figure 1 (a) plots the net RF photocurrent outside the Brillouin resonance ($G=1$) in blue, which corresponds to the filter passband, as a function of DC bias offset from the optimal value of 0.44. Similarly, the net RF photocurrent at the Brillouin resonant frequency ($G > 1$) is in orange, which corresponds to the filter stopband. A significant reduction of the RF output power is observed inside the SBS resonance around $\Delta V_{DC} = 0$ due to the interferometric nature of the MWP notch filtering scheme and corresponds to the RF notch.

Figure 1 (b) plots the RF rejection ratio as a function of ΔV_{DC} , which is calculated by subtracting the RF passband power from the RF stopband power. Large (>60 dB) RF rejection ratio is observed due to the destructively interfering photocurrents at $\Delta V_{DC} = 0$. The black dashed line marks the RF rejection ratio of 51 dB that we experimentally reported in this work. A rejection ratio of 40 dB can be maintained if $|\Delta V_{DC}| < 4.1$ mV. The sensitivity of RF rejection ratio on SBS gain is reduced with larger magnitudes of SBS gain, and hence reduced As_2S_3 waveguide losses. For example, with 5 dB of SBS gain, a rejection ratio of 40 dB can be achieved $|\Delta V_{DC}| < 14.2$ mV.

We note that the sensitivity of RF rejection to bias voltage is not specific to Brillouin-based filters. Many MWP notch filters that use interferometric techniques to enhance RF notch depth have similar sensitivity of RF notch depth to the modulator DC bias voltage.

Figure 1. Simulated RF rejection ratio dependency as a function of DDMZM DC bias voltage. (a) Simulated RF output of MWP filter as a function of DC bias voltage for frequencies that are outside the SBS resonance (blue) and frequencies that correspond to the SBS central frequency (orange). (b) RF Rejection ratio as a function of DC bias voltage.

Pumping Wavelength:

The wavelength of the pump laser determines the frequency of the Brillouin gain resonance. In this experiment, we used lasers with wavelengths around 1550 nm. For the pump wavelength tuning range used in this work (10's GHz), changing the pump wavelength causes the frequency of the RF notch frequency to change by the amount that the Brillouin pump was changed. This is because the BFS changes minimally over the pump wavelength range. For example, the BFS was measured to change at a rate of only 7 MHz/nm in optical fibres [20]. Additionally, there will be negligible change to the magnitude of generated Brillouin gain caused by changes in the pump wavelength [21]. We added the following in Section II C of the updated manuscript to reflect this information:

"The notch frequency was tuned by changing the frequency of the Brillouin pump relative to the probe."

Wider frequency range:

We thank the reviewer for their recommendation to include further experimental demonstrations. At present, the RF notch tuning demonstration was limited in frequency range by the limited magnitude of generated SBS gain and the frequency roll-off of the on-chip modulator.

As discussed in the first question from this reviewer, a significant RF link gain penalty of approximately 20 dB is observed due to strong destructive RF interference from the modest SBS gain generated in current devices. The upper frequency of the RF filter measurement also limited by RF link gain reduction caused by the roll-off in the E-O conversion efficiency of the on-chip DDMZM.

With improved device fabrication process flows in the future, the magnitude of generated Brillouin gain in the As_2S_3 waveguides is expected to increase significantly (from 0.8 dB to 17.8 dB) and would lead to an overall RF link gain improvement of 44.5 dB. This would remove many of the challenges outlined above, enabling an increased notch frequency tuning range of the MWP filter. The following passage reflects these points and was added to Section II C of the updated manuscript:

“The maximum notch frequency is currently limited by the limited magnitude of generated SBS gain and the roll-off of the on-chip E-O modulator response. Additionally, the RF notch depth is sensitive to amplitude and phase perturbations of the optical carrier and sidebands. Such perturbations can arise from changes to the DDMZM bias voltage (See Supplementary Information Section III C). However, with reduced As_2S_3 losses, the magnitude of applied Brillouin gain will increase, enabling a larger notch frequency tuning range and reducing the sensitivity of the RF notch depth to environmental factors.”

Emphasising the advantage of SBS:

We thank the reviewer for their suggestion and insights. We have added the following statement to Section I of the revised manuscript which reflects the advantages SBS being FSR-free. Further details can be found as part of the response to the second comment of this reviewer:

“In addition to fine spectral resolution, SBS offers several advantages for MWP processing applications as the Brillouin response does not exhibit an FSR, which does not impose any limitation on the operating range of the resultant MWP system [1, 2]...”

4. While the characterizations of the microwave photonic filter are based on small signal testing using a VNA, it would be advantageous to include results that demonstrate the suppression of real-world microwave signals, similar to the approach in [Tao, Yuansheng, et al., *Photonics Research* 9.8 (2021): 1569-1580], for verification.

We thank the reviewer for the suggestion. However, here we mainly focussed on the breakthrough of achieving high-level photonic integration. Considering this focus, we prioritised measurements that were critical to characterise the active on-chip E-O components and As_2S_3 waveguides, which were key to the novelty of our work. The demonstrated RF link performance so far is not optimized to perform a filtering demonstration using real-world RF signals. It would be more meaningful to show the real-world signal filtering result with the future improvement of the link performance.

There is proof that Brillouin MWP filters indeed have the performance required to process such RF signals. This has been previously reported by the lead author of this work in Ref. [22] using discrete optical components, which showed a comprehensive demonstration of a Brillouin-based MWP filter suppressing interfering signals, thereby allowing the transmission of real-world data.

5. In the main text, it is important to provide a detailed discussion of the fabrication processes employed to reduce waveguide losses.

We thank the reviewer with their insights that provide us an opportunity to improve the clarity of our work. We have updated the main text of the manuscript in Section II A to clarify the difference in the fabrication process between the standalone heterogeneous Si-As₂S₃ platform, with increased waveguide losses, and the standalone As₂S₃ platform, with reduced waveguide losses.

“One can note that the only difference in the fabrication process between the standalone As₂S₃ chips and the Si-As₂S₃ chips was that the standalone chips were not exposed to HF before As₂S₃ deposition.”

Lower losses were achieved by using a As₂S₃ rib waveguide (680 nm film thickness, 50% etch depth, 1800 nm width) on the integrated Si-As₂S₃ platform (1.71 dB/cm) rather than a fully-etched As₂S₃ waveguide (2.81 dB/cm). Additionally, significantly lower losses were observed on a standalone As₂S₃ chip that was not exposed to hydrofluoric acid (HF) (0.38 dB/cm for both fully-etched and rib). The only difference in fabrication of the standalone and heterogeneously integrated waveguide was that the preconditional wet HF etching step that was not performed on the standalone As₂S₃ chip, which indicates that this additional etching step is a main cause for the increased loss.

Although further characterisation of each parameter used in the fabrication process flow is required, our measurements indicate that the additional losses in the As₂S₃ waveguides in the heterogenous Si-As₂S₃ platform could be due to sidewall scattering and Silica substrate roughness. By using a rib waveguide design, the interactions between the optical mode and sidewall are reduced, thereby lowering losses.

To help improve the logical flow of the manuscript, we have:

- (1) Clarified that the difference in fabrication steps between the standalone As₂S₃ platform (with lower losses) and the Si-As₂S₃ platform was the initial HF etching step before chalcogenide deposition (this change was identified above):

“One can note that the only difference in the fabrication process between the standalone As₂S₃ chips and the Si-As₂S₃ chips was that the standalone chips were not exposed to HF before As₂S₃ deposition.”

- (2) Moved the paragraph that outlines the measured As₂S₃ losses immediately after the discussion of fabrication steps and waveguide geometry. This paragraph is now found in Section II A on Page 4 and starts with

“We measured As₂S₃ propagation losses of 1.7 dB/cm in the Brillouin spiral waveguides...”

- (3) Updated the discussion regarding SiO₂ surface roughness in Section II A of the manuscript to improve clarity:

“Scattering from waveguide sidewalls, as well as surface roughness induced from the HF etching step could have increased the losses of the As_2S_3 waveguides on the heterogeneous platform. This is evidenced by the reduced As_2S_3 propagation losses on the standalone As_2S_3 platform that was not exposed to HF before chalcogenide deposition, as well as the increased losses of the fully-etched waveguides compared to the 50% etched waveguides on the Si- As_2S_3 platform. We measured the root mean squared (RMS) roughness of the SiO_2 substrate on the heterogeneous platform before and after HF etching using atomic force microscopy (AFM). The RMS roughness increased from 0.3 nm to 3.2 nm after HF etching. The increased SiO_2 roughness, in turn, affects the surface roughness of the As_2S_3 film that is deposited in the Si- As_2S_3 platform. This can lead to increased sidewall roughness once the waveguides undergo RIE etching and can explain why the rib waveguide exhibits significantly reduced propagation losses compared to the fully-etched waveguide on the Si- As_2S_3 platform (See Supplementary Information Section I).”

6. Subscript should be used in the title of Fig.1 b

We thank the reviewer for their correction and have updated the figure in the manuscript, which we include below for convenience:

7. Since bi-directional pumping necessitates the use of a circulator, which poses challenges for chip-based integration, it would be valuable to clarify whether this Si- As_2S_3 platform supports forward SBS as an alternative approach.

We acknowledge the reviewer’s comment and note that forward SBS has recently been theoretically predicted in planar single-pass As_2S_3 waveguides, highlighting the possibility of inducing forward SBS in As_2S_3 and As_2Se_3 [23]. However, its experimental realisation remains challenging. It is worth noting that although forward SBS does not require circulators, it places stringent requirements to include narrowband optical filters with large rejection and sharp roll-off to attenuate the high-powered optical pump that is close in frequency and co-propagates with the Brillouin probe. Such a type filter is so far challenging to be implemented on the chip, which is however possible with the future development.

Alternatively, the PPER scheme is a forward Brillouin interaction whereby the pump and probe are in separate waveguides, therefore easing the stringent requirements placed on the optical filter. However, the frequency tunability and reconfigurability of such schemes are limited, as discussed in Ref. [6].

Although it is true that systems that use backward SBS typically require a circulator to facilitate bi-directional pumping, it is possible to induce and harness SBS processing without requiring a circulator. For example, Liu *et al.* demonstrated on-chip backward intermodal SBS in planar As_2S_3 waveguides and used on-chip modal couplers in place of circulators [24], thus simplifying the overall system requirements. An alternate path towards a circulator-free and fully-integrated backward Brillouin MWP processor is depicted in Figure 2. While this architecture circumvents the need for on-chip non-reciprocal devices by using an on-chip tunable coupler (TC3) to separate the counter-propagating pump and probe, it introduces an inherent trade-off between RF link gain and Brillouin pump power based on the tunable coupler power splitting ratio, α . Optical filters, represented by MRR1-3, are used to filter the counter-propagating signals to protect each laser source.

Figure 2 Schematic for a fully-integrated and circulator-free Brillouin circuit. This device leverages on-chip optical filters and a tunable coupler in place of non-reciprocal components to provide a more short-term pathway towards a fully-integrated Brillouin MWP processor. PIC – photonic integrated circuit, MRR: Micro-ring resonator, TC: Tunable coupler, VOA: Variable optical attenuator, TPS: Tunable phase shifter, E-O: Electro-optic, PD: Photodetector.

We added the following to Section III of the updated manuscript to reflect the above points:

“Similarly, incorporating intermodal SBS [24], forwards SBS [60], or using tunable couplers to separate the counterpropagating pump and probe could enable systems where non-reciprocal devices are not needed, simplifying the overall system design.”

8. The utilization of the novel modulation control scheme results in a higher carrier-to-signal ratio. However, the extra link gain is actually provided by the external optical amplifier. It is crucial to consider the influence of additional amplified spontaneous emission (ASE) noise introduced by the external optical amplifier on the filtering performance. The authors should discuss the impact of this noise.

We thank the reviewer for the opportunity to improve the technical content of our manuscript. In this work, the low seed power into the erbium-doped fibre amplifier (EDFA) causes a significant increase in ASE noise when the optical carrier is suppressed. We have added the following discussion on the added ASE noise to Section II C of the manuscript.

“The weak seed power into the EDFA causes significant amplified spontaneous emission (ASE) noise in the EDFA output. The excessive ASE noise is expected to be reduced with lower As_2S_3 losses, such that a net RF link gain and NF improvement is observed with CSR suppression, as observed in Ref. [52].”

It is true that the optical amplifier is required to improve the RF link gain when the novel E-O modulator is used. However, the novel E-O modulator is also a key component that enables RF link gain enhancement through CSR control, which cannot be achieved using a standard intensity modulator.

In fact, Daulay *et al.* used an optical carrier suppression scheme and observed an improvement in RF link gain, as well as a reduction in noise figure as the optical carrier was suppressed [25]. In our work, there is no net RF noise figure improvement due to the presence of excessive ASE noise, despite the use of an optical bandpass filter (BPF3) to reject the broadband ASE noise. We expect that the added ASE noise will be significantly reduced with lower As_2S_3 waveguide losses, due to a higher seed power in to the EDFA.

The broadband ASE noise can be seen in Fig. 5 (e) in the main manuscript. We note that ASE noise exists in the output of all EDFAs regardless of optical input power.

In all, I find this work, although constrained by losses, is still an interesting attempt to integrate the nonlinear functions into the foundry-based active silicon photonic circuits.

We thank the reviewer for their comments and positive feedback on our work. We are convinced that our responses have helped to highlight the novelty of the work we have presented, and the exciting possibilities this novel platform offers for MWP signal processing in a compact footprint.

Reviewer #2 (Remarks to the Author):

In this research article, the authors propose a compact 5mmx 5mm chip-scale microwave photonic notch filter include active E-O components and heterogeneously integrated As₂S₃ waveguides, which can provide a Brillouin gain of 44.5dB to the link. The reported filter achieves a fine spectral resolution and RF rejection of 37MHz and 51dB, respectively. In addition, the proposed filter demonstrates a potential of heterogeneous integration based on silicon photonic platform and Brillouin-based waveguides in implementing on-chip MWP processing functionalities, including phase shifters, true time delays, frequency converters and RF sources. Also, the paper and supplementary material are well written and ideas are well explained with adequate simulations and experimental results to support the claims. Here are my comments and question for the authors:

1. I understand that the Brillouin gain is highly desired to MWP filter, and with the heterogeneous integrations of Brillouin Si-As₂S₃ waveguides, it is possible to achieve a fully integrated MWP system, however, what is the price of implementing such heterogeneous integrations?

We acknowledge the reviewer's comment regarding the price of implementing the system we demonstrated in this work. At present, the silicon photonic chip was fabricated by IMEC foundry using the standard iSiPP50G PDK. Post-processing of As₂S₃ waveguides were completed in university R&D fabrication facilities, which can be done at wafer-scale.

For the current research purpose, the price is approximately \$270 USD per filter (based on the fact of 4 filters on one 5 mm x 5 mm chip). Being a research prototype rather than produced at scale, the current price remains higher than off-the-shelf MEMS filters or electronic filters, however the heterogeneous Brillouin filter in this work enables much wider tunability, and continuous reconfigurability. For the near future, this technology is more suitable for the specialised applications in space or defence where the cost is not the main concern and the functionality enhancement and footprint play a key role. Viewed more broadly, with future optimisation of wafer-scale fabrication process, we predict that the cost per device will be significantly reduced with increasing volume of devices.

2. In order to show the advantage of proposed heterogeneous integration, it is better to give a performance summary or comparison between this work and other reported state-of-art silicon photonic microwave filters.

We thank the reviewer for their suggestion. We have included a table to highlight the benefits of integration of Brillouin devices with active on-chip E-O components.

We have added the following table and discussion to Section III of the updated manuscript to highlight the advantages of this work in enabling a new class of MWP filters that combine fine spectral resolution Brillouin processing with active E-O components on a single compact chip:

“Table I summarises the performance of state-of-the-art MWP filters which exhibit a high degree of integration. It shows that this work is key in unifying the advantages of nonlinear Brillouin signal processing, which offers fine spectral resolution and reconfigurability, with active E-O components. An additional benefit of our work is the flexibility enabled by the novel on-chip E-O modulator, enabling RF link gain improvement through CSR control and simultaneously facilitates the MWP filter to be altered between bandpass and notch responses.

The advantage of this work is further highlighted by comparing the reported 3-dB bandwidth of the MWP filters. Although some MRR and tapped delay line demonstrations have successfully integrated on-chip E-O devices [10, 20], they were generally limited to 3-dB bandwidths that exceeded hundreds of MHz. Additionally, Brillouin processing offers advantages over alternate optical signal processing methods. For example, the FSR of MRRs and tapped delay line MWP filters limit their frequency tunability. However, Brillouin MWP filters do not face such limitations as there is no FSR associated with the SBS process. Furthermore, the thermal stability of the Brillouin frequency shift (≈ 1 MHz/ $^{\circ}$ C in optical fibres [25]) ensures stable notch operation, whereas the frequency of high-Q MRRs often exhibits a strong dependency on temperature (e.g. ≈ 12 GHz/ $^{\circ}$ C was reported for a silicon MRR with Q of 10^7 in Ref. [16]), which introduces a challenging requirement to either stabilise the MRR, or lock the laser to the MRR resonant frequency. Although the 21 MHz 3-dB bandwidth demonstrated by the MRR-based MWP filter in Ref. [16] is promising for future generations of MRR-based systems, the device did not include an E-O interface and faces challenges from thermal sensitivity and the presence of an FSR in its response, as outlined above.”

Table 1 State-of-the-art integrated MWP filters.

Year	Material Platform	RF Filter Response	Optical Processing	On-chip E-O?	FSR-free operation?	On-chip MWP link improvement?	3-dB Bandwidth (MHz)	Ref
2021	InP + Si	Bandpass, Notch	MRR	Y	N	N	360-470	[1]
2022	InP + GaAs + Si	Bandpass	TDL	Y	N	N	900-1, 100	[2]
2022	Si ₃ N ₄	Bandpass, Notch	MRR	N	N	Y	400	[3]
2022	Si	Bandpass	MRR	N	N	N	20.6	[4]
2020	As ₂ S ₃	Bandpass	SBS	N	Y	N	30-350	[5]
2022	Si	Notch	FSBS	N	Y	N	2.7	[6]
2023	Si + As ₂ S ₃	Bandpass, Notch	SBS	Y	Y	Y	37	This Work

MRR - Microring resonator, TDL - Tapped delay line, SBS - Stimulated Brillouin scattering, FSBS - Forward SBS, E-O - Electro-optic

3. How much does the performance of Brillouin Si-As₂S₃ waveguides change with the temperature? Or is there any unique requirement to silicon photonic process to achieve the proposed heterogeneous integration ?

We thank the reviewer's questions which allow us to provide additional information on our work.

Temperature:

It is well known that the Brillouin frequency shift (BFS) changes as a function of temperature. Although we did not characterise BFS of the As_2S_3 waveguides as a function of temperature, the BFS in polarisation maintaining silica fibres has been measured to be only 1.0580 MHz/°C [17]. We note that the sensitivity of the BFS to temperature is much lower than thermally tuned MRRs. For example, the MRR presented in Ref. [4] exhibits a dependence of central frequency to temperature of approximately 12 GHz/°C.

Unique requirements to silicon photonic fabrication process:

This silicon photonic chip itself is fabricated using standard foundry processes. We used custom post-processing steps to augment the bare silicon photonic chip with As_2S_3 Brillouin waveguides. These custom post-processing steps required an opening in the cladding to be made through HF etching. We note that IMEC now offers a fabrication flow whereby the silicon waveguides are exposed and not covered by a silica cladding. This therefore removes the need for a post-processing HF etching step, greatly simplifying the fabrication flow to integrate As_2S_3 waveguides. A lift-off process was used to ensure that only the trench was exposed to HF acid, and to remove the deposited As_2S_3 film outside of the trench region. There are negligible adverse thermal effects to the Si chip during the As_2S_3 post-processing steps, as it was exposed to low temperatures (< 50 °C) that are far below the maximum limit of standard CMOS processing steps (100's °C).

We have added the following in Section II A of the updated manuscript to reflect this:

"Figure 2 presents the details of the As_2S_3 waveguides that, when integrated into the active silicon photonic platform using custom post-processing steps outlined below, enhance the standard silicon photonic chips with nonlinear Brillouin processing capabilities."

4. The lengths of Si- As_2S_3 in supplementary materials (Table S1) and the manuscript (session III) are different, please check the value.

We thank the reviewer for the careful attention paid to the manuscript. Indeed, we stated the length of the As_2S_3 waveguide at 5.6 cm in the manuscript and 5.5 cm in the supplementary materials. The true waveguide length was 5.551 cm. We have corrected the length to **5.56 cm** through the updated manuscript and supplementary material.

5. Is the photodetector in Fig.1 only used for the characterization of E-O modulator? Why do not used it to replace the off-chip PD in Fig.5 to increase the integration density of Si- As_2S_3 MWP notch filter. Is there any other design concern?

We did not use the on-chip PD for characterisation of the MWP notch filter due to the additional RF losses induced by the As_2S_3 waveguides, that were not present of the MWP link-only measurement. This caused the MWP link gain for the filter to be less than the crosstalk between the RF probes used to interface the DDMZM and PD. For more information, please refer to comments 1 and 7 from Reviewer 1 and the corresponding change to the manuscript in Section II C.

“We did not use the PD on the Si- As_2S_3 chip for the MWP filtering demonstration because the MWP filter exhibited increased RF losses compared to the intensity-modulation direct-detection (IM-DD) link-only measurement without the As_2S_3 waveguide. As a result, the MWP link gain of the filter was below the crosstalk between the RF probes that drive the modulator and PD. The main sources of RF loss are attributed to two main factors. The first factor is the increased insertion losses (12.3 dB) of the As_2S_3 waveguides, which is linked to the sample preconditioning removes a $2\ \mu\text{m}$ silica cladding layer to expose the silicon waveguides. The second source of loss is the partial destructive interference of RF photocurrents that are used to generate the MWP notch filter scheme due to the small 0.8 dB of SBS gain generated on the Si- As_2S_3 chip (See Supplementary Information Section III C). These RF losses can feasibly be overcome by improving the As_2S_3 post-processing fabrication flow, reducing sidewall scattering through further design optimisation of As_2S_3 waveguide dimensions, and avoiding the silica cladding etching step by including dedicated open sections on the initial silicon circuit. Reducing the As_2S_3 waveguide losses will improve the link gain through two mechanisms, a direct improvement from lower optical insertion loss and larger Brillouin gain via the increase in effective length.”

6. Is the proposed Si- As_2S_3 waveguide sensitive to the polarization?

Yes, the system is sensitive to polarisation, as the waveguides are non-symmetric. The grating couplers on the silicon photonic chip are optimised to couple light from an optical fibre into the transverse electric (TE) mode of the on-chip waveguide. We ensure that TE polarisation is incident on the chip by adjusting the polarisation such that the output optical power from the chip is maximised. Additionally, SBS is a polarisation-dependent gain process. Because the pump and probe are both aligned to excite the fundamental TE mode, the SBS gain is maximised.

The schematic in Fig. 4 (a) of the original manuscript was missing a polarisation controller. We acknowledge this could have caused some confusion. In the revised manuscript, we have updated Fig. 4 (a) such that there is now a polarisation controller (PC1) between the input laser and the chip. The updated figure is presented below for convenience:

We have updated the manuscript with the following statement in Section II A to acknowledge the polarisation dependence of the waveguides in the heterogenous Si-As₂S₃ platform:

“We ensured that light was coupled to the TE₀₀ mode of this silicon waveguide by adjusting the polarisation of the input light into the chip until the transmitted power was maximised.”

References:

- [1] Y. Tao *et al.*, ‘Hybrid-integrated high-performance microwave photonic filter with switchable response’, *Photonics Research*, vol. 9, no. 8, p. 1569, Aug. 2021, doi: 10.1364/PRJ.427393.
- [2] H. Shu *et al.*, ‘Microcomb-driven silicon photonic systems’, *Nature*, vol. 605, no. 7910, pp. 457–463, 2022, doi: 10.1038/s41586-022-04579-3.
- [3] O. Daulay *et al.*, ‘Ultrahigh dynamic range and low noise figure programmable integrated microwave photonic filter’, *Nat Commun*, vol. 13, no. 1, p. 7798, Dec. 2022, doi: 10.1038/s41467-022-35485-x.
- [4] L. Zhang *et al.*, ‘Ultralow-Loss Silicon Photonics beyond the Singlemode Regime’, *Laser & Photonics Reviews*, vol. 16, p. 2100292, Jan. 2022, doi: 10.1002/lpor.202100292.
- [5] M. Garrett, Y. Liu, P. Ma, D.-Y. Choi, S. J. Madden, and B. J. Eggleton, ‘Low-RF-loss and large-rejection reconfigurable Brillouin-based RF photonic bandpass filter’, *Optics Letters*, vol. 45, no. 13, p. 3705, 2020, doi: 10.1364/ol.395477.
- [6] S. Gertler *et al.*, ‘Narrowband microwave-photonic notch filters using Brillouin-based signal transduction in silicon’, *Nature Communications*, vol. 13, no. 1, p. 1947, Dec. 2022, doi: 10.1038/s41467-022-29590-0.
- [7] B. Haq *et al.*, ‘Micro-Transfer-Printed III-V-on-Silicon C-Band Semiconductor Optical Amplifiers’, *Laser and Photonics Reviews*, vol. 14, no. 7, pp. 1–12, 2020, doi: 10.1002/lpor.201900364.
- [8] K. Van Gasse, R. Wang, and G. Roelkens, ‘27 dB gain III–V-on-silicon semiconductor optical amplifier with > 17 dBm output power’, *Optics Express*, vol. 27, no. 1, p. 293, 2019, doi: 10.1364/oe.27.000293.
- [9] Y. Liu *et al.*, ‘A photonic integrated circuit-based erbium-doped amplifier’, *Science*, vol. 376, no. 6599, pp. 1309–1313, Jun. 2022, doi: 10.1126/science.abo2631.
- [10] S. Gertler, E. A. Kittlaus, N. T. Otterstrom, and P. T. Rakich, ‘Tunable microwave-photonic filtering with high out-of-band rejection in silicon’, *APL Photonics*, vol. 5, no. 9, pp. 1–30, 2020, doi: 10.1063/5.0015174.

- [11] W. Xu, A. Iyer, L. Jin, S. Y. Set, and W. H. Renninger, 'Strong optomechanical interactions with long-lived fundamental acoustic waves', *Optica*, *OPTICA*, vol. 10, no. 2, pp. 206–213, Feb. 2023, doi: 10.1364/OPTICA.476764.
- [12] Y. Dong *et al.*, 'Sub-MHz ultrahigh-resolution optical spectrometry based on Brillouin dynamic gratings', *Opt. Lett.*, vol. 39, no. 10, p. 2967, May 2014, doi: 10.1364/OL.39.002967.
- [13] S. Preußler, A. Wiatrek, K. Jamshidi, and T. Schneider, 'Brillouin scattering gain bandwidth reduction down to 3.4MHz', *Opt. Express*, vol. 19, no. 9, p. 8565, Apr. 2011, doi: 10.1364/OE.19.008565.
- [14] A. L. Gaeta and R. W. Boyd, 'Stochastic dynamics of stimulated Brillouin scattering in an optical fiber', *Phys. Rev. A*, vol. 44, no. 5, pp. 3205–3209, Sep. 1991, doi: 10.1103/PhysRevA.44.3205.
- [15] A. Choudhary *et al.*, 'Advanced Integrated Microwave Signal Processing with Giant On-Chip Brillouin Gain', *Journal of Lightwave Technology*, vol. 35, no. 4, pp. 846–854, 2017, doi: 10.1109/JLT.2016.2613558.
- [16] D. Marpaung *et al.*, 'Low-power, chip-based stimulated Brillouin scattering microwave photonic filter with ultrahigh selectivity', *Optica*, vol. 2, no. 2, p. 76, 2015, doi: 10.1364/optica.2.000076.
- [17] W. Zou, Z. He, and K. Hotate, 'Complete discrimination of strain and temperature using Brillouin frequency shift and birefringence in a polarization-maintaining fiber', *Opt. Express*, vol. 17, no. 3, p. 1248, Feb. 2009, doi: 10.1364/OE.17.001248.
- [18] E. Grafer, G. Miles, A. Cramer, A. Klee, C. Middleton, and R. Desalvo, 'RF Photonic Equalization Using Stimulated Brillouin Scattering', *Proceedings - IEEE Military Communications Conference MILCOM*, vol. 2019-October, pp. 893–895, 2019, doi: 10.1109/MILCOM.2018.8599717.
- [19] A. Zadok, A. Eyal, and M. Tur, 'Gigahertz-wide optically reconfigurable filters using stimulated Brillouin scattering', *Journal of Lightwave Technology*, vol. 25, no. 8, pp. 2168–2174, 2007, doi: 10.1109/JLT.2007.901336.
- [20] A. Zarifi, M. Merklein, Y. Liu, A. Choudhary, B. J. Eggleton, and B. Corcoran, 'Wide-range optical carrier recovery via broadened Brillouin filters', *Opt. Lett., OL*, vol. 46, no. 2, pp. 166–169, Jan. 2021, doi: 10.1364/OL.411482.
- [21] B. J. Eggleton, C. G. Poulton, and R. Pant, 'Inducing and harnessing stimulated Brillouin scattering in photonic integrated circuits', *Advances in Optics and Photonics*, vol. 5, no. 4, p. 536, Dec. 2013, doi: 10.1364/AOP.5.000536.
- [22] M. Garrett *et al.*, 'Multi-Band and Frequency-Agile Chip-Based RF Photonic Filter for Ultra-Deep Interference Rejection', *Journal of Lightwave Technology*, vol. 40, no. 6, pp. 1672–1680, Mar. 2022, doi: 10.1109/JLT.2021.3129509.
- [23] C. K. Lai *et al.*, 'Optimizing performance for an on-chip stimulated Brillouin scattering-based isolator', *J. Opt. Soc. Am. B*, vol. 40, no. 3, p. 523, Mar. 2023, doi: 10.1364/JOSAB.479629.
- [24] Y. Liu *et al.*, 'Circulator-Free Brillouin Photonic Planar Circuit', *Laser and Photonics Reviews*, vol. 2000481, pp. 1–7, 2021, doi: 10.1002/lpor.202000481.
- [25] O. Daulay, G. Liu, and D. Marpaung, 'Microwave photonic notch filter with integrated phase-to-intensity modulation transformation and optical carrier suppression', *Optics Letters*, vol. 46, no. 3, p. 488, 2021, doi: 10.1364/ol.413579.

REVIEWERS' COMMENTS

Reviewer #1 (Remarks to the Author):

The author has addressed all my concerns. I therefore recommend for publication of this manuscript.

Reviewer #2 (Remarks to the Author):

1. I believe the authors didn't properly address my comments regarding "price". I mean is there any other design constrain or performance difference caused by such heterogeneous integration.

2. Can please provide more analysis about implementing this Brillouin-based waveguides to other application area? For example, is it possible to replace EDFA by this heterogeneous integration in an optical communication system?

Except the above questions, the authors have addressed all my concerns.

REVIEWERS' COMMENTS

Reviewer #1 (Remarks to the Author):

The author has addressed all my concerns. I therefore recommend for publication of this manuscript. We thank the reviewer for the endorsement of our work.

Reviewer #2 (Remarks to the Author):

1. I believe the authors didn't properly address my comments regarding "price". I mean is there any other design constrain or performance difference caused by such heterogeneous integration.

We apologise for our misunderstanding of the reviewer's question related to the price. Regarding performance differences caused by the heterogeneous integration, the custom post-processing steps used in this proof-of-concept demonstration required to integrate the As_2S_3 waveguide caused additional losses and reduced the magnitude of the generated SBS gain. The losses are predominantly attributed to the etching of the silica cladding required in the current demonstration. This constrained the design by restricting the simultaneous use of the on-chip PD and modulator with the As_2S_3 waveguides. Please refer to Section II A for a detailed discussion of As_2S_3 waveguide losses.

The above limitations can be overcome in future implementations of this device, such as by including an opening that fully exposes the silicon photonic waveguide. Please refer to Section III of the updated manuscript for a discussion on techniques to improve performance in the future, as well as an analysis of the effect of the increased As_2S_3 waveguide losses on MWP filter performance and the magnitude of generated SBS gain.

2. Can please provide more analysis about implementing this Brillouin-based waveguides to other application area? For example, is it possible to replace EDFA by this heterogeneous integration in an optical communication system?

Brillouin-based photonic systems can be utilized in many application areas in addition to microwave photonics (MWP), and the platform presented in this work shows a pathway toward integrating Brillouin-based systems on a single photonic chip. Some potential applications are outlined below.

The Brillouin-based chip that we present in this work could be used to implement an SBS-based sensor, whereby the physical properties of a material, including strain and temperature, can be determined from the properties (linewidth, Brillouin frequency shift) of the Brillouin response [1], [2]. Although Brillouin sensors have been implemented using chip-scale components [3], similar to MWP applications, these demonstrations rely on off-chip electro-optic components. Therefore, the presented heterogeneous integration platform could be used for sensing applications.

The heterogeneous platform we present in this work can also be applied to perform optical carrier recovery in high-speed communications systems [1], [4], [5] and provide significant footprint reduction by integrating E-O components on the same platform as the Brillouin gain medium.

Within the MWP community, Brillouin-based systems are highly desirable, owing to the narrow linewidth and high degree of reconfigurability offered by the Brillouin response. As a result, many key building blocks of microwave systems, beyond the MWP filter shown in this manuscript, have been demonstrated using Brillouin scattering, including phase-shifters [6], [7], true time delays [8], [9], frequency converters [10], [11] and microwave sources [12], [13]. All those demonstrations would benefit from the significant footprint reduction offered by heterogeneous integration. For a detailed outline of the use of on-chip SBS in MWP applications, we direct the reviewer to Ref. [14].

Finally, it could indeed be possible for this system to replace an EDFA in an optical communication system to amplify narrowband signals, such as for optical carrier recovery, as mentioned above [4], [5]. It has been demonstrated that the frequency-selective Brillouin gain applied to the carrier can achieve a performance advantage compared to broadband amplification from an EDFA. The heterogeneously integrated platform demonstrated in this paper would greatly reduce the footprint of such a system and offer potential advantages regarding stability, particularly when compared to fiber optic based Brillouin schemes.

Except the above questions, the authors have addressed all my concerns.

We thank the reviewer for their positive reception of our work.

References

- [1] B. J. Eggleton, M. J. Steel, and C. Poulton, Eds., *Brillouin scattering. Part 2*, First edition. in Semiconductors and semimetals, no. volume 110. Cambridge, MA San Diego, CA Kidlington, Oxford London: Academic Press, an imprint of Elsevier, 2022.
- [2] M. Merklein, I. V. Kabakova, A. Zarifi, and B. J. Eggleton, '100 years of Brillouin scattering: Historical and future perspectives', *Applied Physics Reviews*, vol. 9, no. 4, p. 041306, 2022, doi: 10.1063/5.0095488.
- [3] A. Zarifi *et al.*, 'Highly localized distributed Brillouin scattering response in a photonic integrated circuit', *APL Photonics*, vol. 3, no. 3, p. 036101, Mar. 2018, doi: 10.1063/1.5000108.
- [4] A. Zarifi, M. Merklein, Y. Liu, A. Choudhary, B. J. Eggleton, and B. Corcoran, 'Wide-range optical carrier recovery via broadened Brillouin filters', *Opt. Lett., OL*, vol. 46, no. 2, pp. 166–169, Jan. 2021, doi: 10.1364/OL.411482.
- [5] E. Giacomidis *et al.*, 'Chip-based Brillouin processing for carrier recovery in self-coherent optical communications', *Optica, OPTICA*, vol. 5, no. 10, pp. 1191–1199, Oct. 2018, doi: 10.1364/OPTICA.5.001191.
- [6] M. Pagani, D. Marpaung, and B. J. Eggleton, 'Ultra-wideband microwave photonic phase shifter with configurable amplitude response', *Optics Letters*, vol. 39, no. 20, p. 5854, 2014, doi: 10.1364/ol.39.005854.
- [7] L. McKay *et al.*, 'Broadband Brillouin-based Phase Shifter with Phase Amplification in a Silicon Waveguide', *Optica*, vol. 6, no. 7, 2019, doi: <https://doi.org/10.1364/OPTICA.6.000907>.
- [8] I. Aryanfar *et al.*, 'Chip-based Brillouin radio frequency photonic phase shifter and wideband time delay', *Optics Letters*, vol. 42, no. 7, p. 1313, 2017, doi: 10.1364/ol.42.001313.
- [9] L. McKay *et al.*, 'Integrated microwave photonic true-time delay with interferometric delay enhancement based on Brillouin scattering and microring resonators', *Optics Express*, vol. 28, no. 24, p. 36020, Nov. 2020, doi: 10.1364/OE.408617.
- [10] Z. Zhu, D.-Y. Choi, S. J. Madden, B. J. Eggleton, and M. Merklein, 'High-conversion-gain and deep-image-rejection Brillouin chip-based photonic RF mixer', *Optics Letters*, vol. 45, no. 19, p. 5571, 2020, doi: 10.1364/ol.400511.
- [11] L. McKay *et al.*, 'Chip-based SBS for image rejection in a broadband microwave photonic mixer', *Opt. Express*, vol. 31, no. 3, p. 4268, Jan. 2023, doi: 10.1364/OE.482871.
- [12] J. Li, H. Lee, and K. J. Vahala, 'Microwave synthesizer using an on-chip Brillouin oscillator', *Nature Communications*, vol. 4, no. 1, p. 2097, Oct. 2013, doi: 10.1038/ncomms3097.
- [13] M. Merklein *et al.*, 'Widely tunable, low phase noise microwave source based on a photonic chip', *Optics Letters*, vol. 41, no. 20, p. 4633, 2016, doi: 10.1364/ol.41.004633.
- [14] M. Garrett, M. Merklein, and B. J. Eggleton, 'Chip-Based Brillouin Processing for Microwave Photonic Phased Array Antennas', *IEEE Journal of Selected Topics in Quantum Electronics*, vol. 29, no. 1, pp. 1–20, 2023, doi: 10.1109/JSTQE.2022.3197766.